**Trends in heat and cold wave risks for the Italian Trentino Alto-Adige region from 1980 to 2018**

Martin Morlot[1], Simone Russo[2], Luc Feyen[2], and Giuseppe Formetta[1]

[1] University of Trento, Department of civil, environmental, and mechanical engineering,

5 via Mesiano, 77, 38123, Trento (Italy)

[2] European Commission, Joint Research Centre, Via Enrico Fermi, 2749, 21027 Ispra

(Italy)

Corresponding author: Giuseppe Formetta, giuseppe.formetta@unitn.it

**Abstract**

10 Heat waves (HWs) and cold waves (CWs) can have considerable impact on people.

Mapping risks of extreme temperature at local scale, accounting for the interactions

between hazard, exposure and vulnerability, remains a challenging task. In this study,

we quantify risks from HWs and CWs for the Trentino-Alto Adige region of Italy from

1980 to 2018 at high spatial resolution. We use the Heat Wave Magnitude Index daily

15 (HWMId) and the Cold Wave Magnitude Index daily (CWMId) as the hazard indicators.

To obtain HWs and CWs risk maps we combined: i) occurrence probability maps of the

hazard obtained using the zero-inflated Tweedie distribution (accounting directly for the

absence of events for certain years); ii) normalized population density maps; and iii)

normalized vulnerability maps based on eight socioeconomic indicators. The

20 methodology allowed us to disentangle the contributions of each component of the risk

relative to total change in risk. We find a statistically significant increase in HWs hazard

and exposure while CWs hazard remained stagnant in the analyzed area over the study

period. A decrease in vulnerability to extreme temperature spells is observed trough the region except in the larger cities where vulnerability increased. HWs risk increased in 40% of the region, with the increase being greatest in highly populated areas. Stagnant CWs hazard and declining vulnerability result in reduced CWs risk levels overall, except for the four main cities where increased vulnerability and exposure increased risk levels. These findings can help to steer investments in local risk mitigation, and this method can potentially be applied to other regions where there is sufficient detailed data.

## 1   Introduction

Heat waves (HWs) and cold waves (CWs) are hazards that affect public health and the environment (Gasparrini et al., 2015; Habeeb et al., 2015). With global warming, HW intensities and durations are expected to increase while those of CWs are expected to decrease (Perkins-Kirkpatrick and Gibson, 2017; Russo et al., 2015; Smid et al., 2019), changing the risks they pose to society. A recent report showed that in the year 2018 worldwide, 157 million more people were exposed to HWs compared to the year 2000 (Watts et al., 2018). In Europe, recent high intensity HW events (2003 and 2018) -- where HWs are defined as 3 days over $90^{th}$ temperature percentile of the 1980-2010 -- have impacted as much as 55% of its area (García-León et al., 2021). In Italy, HWs had a strong impact on mortality. For example, in 2003, a 27% mortality increase was reported over August compared to August 2002; there was also a 23% increase in July 2015 compared to the same month for the 5 previous years (Michelozzi et al., 2005, 2016). In Trentino Alto-Adige (our study region), Conti et al. (2005) showed that the large HW of 2003, compared to the previous year, increased mortality by 32% in Trento and 28% in Bolzano (the region's two main cities). In the city of Bolzano, it was found

that higher hospital admissions occurred during HW events, particularly among elderly

women (Papathoma-Köhle et al., 2014). With regards to CWs in Europe, recent winters

have claimed lives with 790 deaths in 2006, and 549 deaths in 2012 (Kron et al., 2019).

In Italy, de'Donato et al., (2013) report an increase in mortality (47%) for the timeframe

of the 2012 CW in the city of Bolzano compared to the 4 previous winters (2008-2011).

HWs and CWs events clearly drive risk but how do we define this risk? The United

Nations Office for Disaster Risk Reduction (UNDRR, 2021) and the Intergovernmental

Panel on Climate change (IPCC, 2014) define risk as a function of hazard, exposure,

and vulnerability. Hazard is defined as a process, phenomenon or human activity that

may cause loss of life, injury or other health impacts, property damage, social and

economic disruption or environmental degradation and hazards being characterized by

location, intensity or magnitude, frequency, and probability. Exposure is defined as

people, infrastructure, housing, production, and other tangible human assets present in

hazard-prone areas. Vulnerability is defined as the conditions that define the

susceptibility of an individual, infrastructure, or a community to be impacted by the

hazard. To successfully quantify risk, one must measure all three components: hazard,

exposure, and vulnerability.

With regard to temperature-related hazard and exposure, several studies have been

conducted at global-scale (e.g. Chambers, 2020; Dosio et al., 2018), continental (eg.

King et al., 2018), and at city-scale (e.g. Smid et al., 2019). Most studies focus on

human exposure (eg. Chambers, 2020; Tuholske et al., 2021) and on the exposure of

different land areas (e.g., Ceccherini et al., 2017; Oldenborgh et al., 2019; Russo et al.,

2016). These studies find increasing trends in HWs (Chambers, 2020; Dosio et al.,

2018) and decreasing trends in CWs in their period of analysis (Oldenborgh et al., 2019,

Smid et al., 2019).

 Studies on HWs and CWs typically have used subjective numerical thresholds, on the indicator to define severity and exposure to the hazards (e.g. $0 < HWMId < 3$, $3 < HWMId < 6$, $6 < HWMId < 9$). However, extreme events are usually defined by their return periods. In the case of HWs and CWs, fitting extreme value distributions to define

the return periods is difficult due to the possible absence of events in the analyzed time frame (i.e. zero values, in the case where there are no HWs/CWs in a given year). Generalized extreme value distribution (GEV) and non-stationary-techniques (Dosio et al., 2018; Kishore et al., 2022; Russo et al., 2019) have enabled estimation of HWs and CWs' return periods, but neither approach explicitly accounts for a zero presence in an

analyzed time series.

In this study, for the first time, we use a distribution allowing for the direct fitting of zero-values for extremes (years with no event): the zero-inflated distribution of Tweedie families (Jorgensen, 1987; Tweedie, 1984). This distribution is also used to estimate HWs and CWs frequency of occurrence. The Tweedie distribution has been used

mostly for the purpose of insurance claims analysis. It has seldom been applied in the field of natural hazards, such as HWs mortality (Kim et al., 2017), droughts (Tijdeman et al., 2020), or rainfall analysis (Dunn, 2004; Hasan and Dunn, 2011). The main advantage of the Tweedie distribution is the possibility of considering a range of distributions to describe continuous and semi-continuous domains; these include:

normal; Gamma, Poisson; Compound Gamma-Poisson; and Inverse Gaussian (Bonat and Kokonendji, 2017; Rahma and Kokonendji, 2021; Shono, 2008; Temple, 2018).

Moreover, for some of these distributions (i.e. Poisson mixtures of gamma distributions), the Tweedie distribution approach explicitly enables the fitting of zero-inflated data. The distribution's main limitation is the complex distribution's fitting methodology and the

difficulties in obtaining relevant information criteria, such as the Akaike's information criterion (Shono, 2008) The implication of these limitations are that the 'fitting' of the Tweedie distribution is computationally intensive and that it is difficult to compare its goodness of fit to other distribution via the information criteria.

To perform any risk analysis, vulnerability to the hazard must be quantified. HW and

CW vulnerabilities can be approximated though the combinations of several socioeconomic indicators. Cheng et al. (2021) provide an overview of the different types of indicators used in the literature to quantify vulnerability. The indicators can be diverse, ranging from population structure (e.g., age and health characteristics), social status, economic conditions, community (cultural) group characteristics, and household

physical characteristics. At the community level in the United States, indicators such as social isolation, presence of air conditioning, proportion of elderly and proportion of diabetics in the population have been found to be key for human vulnerability to temperature extremes (Reid et al., 2009). At the national level in Korea, Kim et al. (2017) found that elderly living alone, agricultural workers and unemployed are the main

indicators for vulnerability to heat wave days and tropical nights. Vulnerability indicators, in combination with temperature-mortality relationships, have also been appraised at city scale for HWs (Ellena et al., 2020) and at regional scale (López-Bueno et al., 2021) for CWs (Karanja & Kiage 2021). A study on social vulnerability to natural hazards in Italy (Frigerio and De Amicis, 2016) used 7 indicators (i.e. family structure, education,

socioeconomic status, employment, age, race and ethnicity and population growth) derived from the freely-available census records.

HWs and CWs risks overall are often assessed using different methodologies depending on the objectives of the study. On a global scale, Russo et al., (2019) establish a risk index using the probabilities of HWs as hazard, where the exposure is

the population density normalized in [0;1] based on its maximum, minimum values; while vulnerability is based on a socio-economic indicator (human development index). For Italy, Morabito et al (2015) conducted a risk analysis of heat on elderly in the major cities, using the elderly population as the only vulnerability factor and summer average temperatures for the period 2000-2013 to quantify hazards.

In this study, we assess risk associated with extreme temperatures in the Italian Trentino Alto-Adige region. This is a relevant social and scientific objective given: i) the increase in the percentage of elderly people (i.e. vulnerability change) (Papathoma-Köhle et al., 2014) and ii) changing temperature extremes in view of climate change (i.e. changing hazard). Few studies have attempted to quantify HWs and CWs impacts for

the cities of Trento and Bolzano (main cities of the region), including Conti et al. (2005) as part of their studies on Italian cities and Papathoma-Köhle et al. (2014) who studied impacts in Bolzano. The former compared mortality data of the year 2003, when there was  a very intense HW, to the year 2002, finding an increase of mortality in both Trento and Bolzano. The latter compared hospital admissions due to HWs in summer months

of three years (2003, 2006, and 2009) and found heat health-related issues driving admissions among elderly women.

To understand the evolution of HWs and CWs human risk and to plan adequate risk-mitigation measures in the region of study, the risk and its change at high spatial and temporal resolution need to be analyzed. The aim of this research is to improve

quantification of HW and CW hazards, human exposure, vulnerabilities, over the period 1980-2018, for the Trentino-Alto-Adige region to better assess related risks at high-definition (i.e. city-scale). The goals for this paper are therefore as follows:

1) Quantify HWs and CWs hazards and their return level at a very high spatial resolution (250m) by combining for the first time i) the indicators proposed (HWMId,

CWMId) by Russo et al., (2015) and Smid et al., (2019), together with ii) the Tweedie distribution;

2) Quantify human exposures and vulnerabilities to HWs and CWs and their evolution over time for the Trentino-Alto-Adige region;

3) Quantify HW and CW risks across the region and understand their main drivers,

disentangling how their individual components drive these risks over time.

## 2 Study Area

The Trentino Alto-Adige region (Figure 1) is a mountainous region in northern Italy, which borders Austria. The elevation of the region varies from 65m for lake Garda to 3,905m for the Ortler. It is composed of two provinces (Province of Trento and Province

of Bolzano). Its most populous cities (population for 2022 in parenthesis) are the two provincial capitals, Trento (118509) and Bolzano (107025), as well as minor cities such as Merano (40994) and Rovereto (39819). The main rivers in the region are the Adige, and its tributary, the Isarco. Due to its diverse geography, the climate is also diverse,

ranging from Subcontinental to Alpine on the Koppen classification (Fratianni and

Acquaotta, 2017).

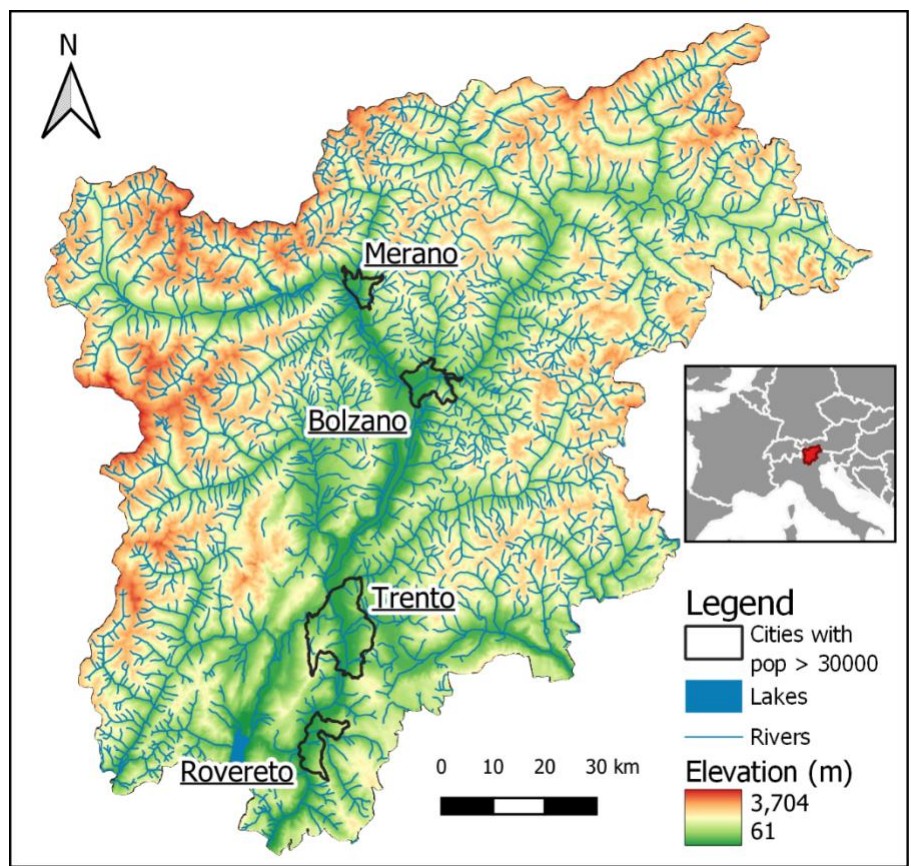

Figure 1: The Trentino Alto-Adige region and its most populated cities (Trento, Bolzano,

Rovereto and Merano); the colors indicating the elevations, river network, and lakes.

## 3    Methodology

### 3.1    Temperature data

In order to quantify the HWs and CW hazard, we used the freely available spatial

temporal temperature dataset by Crespi et al. (2021). It consists of gridded daily

temperatures for the entire Trentino Alto-Adige region covering the period of 1980-2018

at a resolution of 250 meters. The dataset is based on more than 200 station's daily

records that have been quality controlled and homogenized. The interpolation method is

based on a combination of 30-year temperature climatology (1981–2010), daily

anomalies and accounts explicitly for topographic features (i.e. elevation, slope) that are

crucial in orographically complex areas like the Trentino Alto-Adige region. The leave

one out cross validation presented in Crespi et al. (2021) finds a mean correlation

coefficient that is higher than 0.8 and mean absolute errors of around 1.5 degree

Celsius (on average across months and stations used for the interpolation).

### 3.2    Hazard quantification and distribution fitting

### 3.2.1    Hazard quantification

To quantify the hazard, we used the HWMId (Russo et al., 2015) and the CWMId (Smid

et al., 2019). These indices represent a way of measuring extreme temperature events

while considering their durations, intensities, and accounting for site-specific historical

climatology (30years).

According to Russo et al. (2015), HWMId is defined as the maximum magnitude of the

HWs in a year. A HW occurs when the air temperature is above a daily threshold for

more than three consecutive days. The threshold is set to the 90[th] percentile of the

temperature data of the day and the window of 15 days before and after throughout the

reference period 1981-2010. The magnitude of a HW is the sum of the daily heat

magnitude $HM_d$ of all the consecutive days composing the HW (Equation 1):

$$HM_d(T_d) = \begin{cases} \dfrac{T_d - T_{30y25p}}{T_{30y75p} - T_{30y25p}} & \text{if } T_d > T_{30y25p} \\ 0 & \text{if } T_d \leq T_{30y25p} \end{cases}$$

190                                                                                                          (1)

where $HM_d(T_d)$ corresponds to the daily heat magnitude, $T_d$ the temperature of the day in question and $T_{30y25p}$ and $T_{30y75p}$ correspond to the 25th and 75th percentile of the yearly maximum temperature for the 30 years of the reference period (1981-2010). The interquartile range (IQR, i.e. the difference between the $T_{30y75p}$ and $T_{30y25p}$ percentiles of the daily temperature) is used as the heatwave magnitude unit and represents a non-parametric measure of the variability of the temperature timeseries. Therefore, a value of $HM_d$ equals to 3 means that the temperature anomaly on day d with respect to $T_{30y25p}$ is 3 times the IQR. Finally, for a given year HWMId corresponds to the highest sum of magnitude (HMd) over the consecutive days composing a heatwave event (with only days with HMd > 0 considered).

Analogously to the HWMId, CWMId is defined as the minimum magnitude of the CWs in a year (Smid et al., 2019). A CW occurs when the air temperature is below a daily threshold for more than three consecutive days. The threshold is set to the 10th percentile of the temperature data of the day and the window of 15 days before and after throughout the reference period 1981-2010.

The daily cold magnitude corresponds to (Equation 2):

$$CM_d(T_d) = \begin{cases} \dfrac{T_d - T_{30y75p}}{T_{30y75p} - T_{30y25p}} & \text{if } T_d < T_{30y75p} \\ 0 & \text{if } T_d > T_{30y75p} \end{cases}$$

(2)

where $CM_d(T_d)$ corresponds to the cold daily magnitude, $T_d$ the daily temperature and $T_{30y25p}$ and $T_{30y75p}$ correspond to the 25th and 75th percentile yearly temperature for the 30 years used as a reference. Inversely to HWMId, the lowest cumulative magnitude

sum is retained for each year and with only consecutive days with $CM_d < 0$ considered to calculate it. CWMId being always < 0, its absolute values are retained for its values to be on a positive interval (similar to HWMId).

**3.2.2 Distribution fitting**

The HWMId and CWMId yearly values are fitted with a probability distribution to estimate their return periods. Considering that HWMId and CWMId are both defined in [0,+Inf[ , we use the Tweedie distribution (Jorgensen, 1987; Tweedie, 1984), a distribution that can act as zero-inflated, thus accounting for the presence of zeros directly. The Tweedie distribution is an exponential dispersion model which has a probability density function of the form (Equation 3):

$$f(y, \theta, \Phi) = a(y, \Phi) * \exp\left[\frac{1}{\Phi}\{y\theta - \kappa(\theta)\}\right]$$

(3)

where $\Phi$ corresponds to its dispersion parameter that is positive, $\theta$ to its canonical parameter, and $\kappa(\theta)$ the cumulant function. The function a(y, $\Phi$) generally cannot be written in closed form. The cumulant function is related to the mean ($\mu_y = \kappa'(\theta)$) and variance ($\sigma_y = \Phi * \kappa''(\theta)$) and in the case of a Tweedie distribution the variance has a power relationship with the mean (Equation 4):

$$\sigma_y = \Phi * (\mu_y)^p$$

(4)

where p corresponds to the power parameter that is positive.

Depending on the value of p, the distribution will behave differently. In the case where p

is between 1 and 2, it belongs to the compound Poisson-gamma distribution with a

mass at zero, while other p values can make the distribution correspond to a normal,

Poisson, or gamma distribution, among others. The use of the Tweedie distribution is

retained, permitting us to consider the zero values, while also considering other

distributions should there be an absence of zero values.

We fit the distribution to the previously found HWMId and CWMId values with the help

of the Tweedie R package (Dunn, 2021). It provides distribution density, distribution

function, quantile function, random generation for the Tweedie distributions. The

Tweedie parameters (i.e. mean, power, and dispersion) have been estimated by the

"tweedie.profile" function (Dunn, 2021) using the maximum likelihood as described by

Dunn and Smyth (2005). An example of the fitted distribution for Bolzano and Trento

can be found in the supplementary material (Figure S1). It is also possible to use the

same package to estimate a quantile using the fitted distribution, permitting us to

estimate specific return levels for return periods T for both HWMId and CWMId. For this

study two return levels are retained, 5 years (HW5Y for HW, and CW5Y for CW) and 10

years (HW10Y for HWs and CW10Y for CW). This choice aims to account for both the

length of the analyzed period (39 years) and the type of hazards we are analyzing (HWs

and CWs usually do not occur every year). Higher return level estimations would be

affected by extrapolation effects and higher uncertainties.

For statistical fit verification, the Kolmogorov–Smirnov (KS) test on two samples is used

with one sample being the HWMId or CWMId values, and the other sample being a

randomly generated sample using the fitted distribution value. This goodness of fit test

is one of the most commonly used in the literature for zero inflated Tweedie distribution

(Goffard et al., 2019; Johnson et al., 2015; Rahma and Kokonendji, 2021). The null

hypothesis of this test is that the two samples belong to the same distribution. If the P-

value for this test is below the significance level α of 5%, the null hypothesis is rejected,

otherwise we cannot reject the null hypothesis at this significance level.

**3.3   Exposure quantification**

To quantify the population exposed to HWs and CWs, we use time-varying population

data from the Global Human Settlement Layer (GHSL) (Schiavina et al., 2019). The

population data is available at a resolution of 250m for the following years: 1975, 1990,

2000 and 2015. Both these data, and the population count done by the Italian national

statistical institute, indicate a growing population throughout the region in the period for

which data is available(overall 23%, 1975-2015).

To model more accurately exposure, we created yearly varying population maps for the

period 1980-2018, following the methodology presented in other studies (e.g. Formetta

and Feyen, 2019; Neumayer and Barthel, 2011). We linearly interpolated the data in

time for the period 1980 to 2015 (assuming a constant rate in between available years)

and we used the closest year for the period 2016-2018.

Following recent studies (King and Harrington, 2018; Russo et al., 2019), for each year,

a pixel is considered exposed to HW/CWs hazard (or to a 5 or 10 year return-period

HWs/CWs) if, for that year, the HWMId/CWMId of the pixel is greater than zero (or

greater than the corresponding return level HW5Y/CW5Y or HW10Y/CW10Y,

respectively).This is the exposition factor, and it is a binary value (0 meaning un-

exposed or 1 meaning exposed).

The percentage of population exposed is calculated on annual basis over the study

period (1980-2018) and with the help of population datalinearly interpolated from 1980

to 2018.

Using this population data, the percentage of population exposed is then calculated

using the following equations (Equations 5 and 6):

$$Population\ exposed(t) = \sum_i EF_i * population_i(t)$$

(5)

$$Percentage\ of\ population\ exposed\ (t) = \frac{Population\ exposed(t)}{Total\ population\ (t)}$$

(6)

where i corresponds to the pixels, t to the year being analyzed, EF to the exposition

factor mentioned above (binary).

### 3.4   Vulnerability quantification

We express HWs and CWs vulnerability using eight indicators as in Ho et al. (2018);

they quantify community vulnerability to HWs and CWs events based on extreme age,

household physical characteristics, social status and economic conditions. The list of

variables considered is reported in Table 1.

Table 1: Vulnerability indicators used (after Ho et al., 2018)

| Category | Indicator | Definition |
|---|---|---|
| Extreme Age | Older Age | Population over 55 years old |
| | Infants | Population under 5 years old |
| Household physical characteristics | People in old houses | Percentage of household living in housing built prior to 1960 (corresponding to when better insulation started being implemented) |
| | People in poor living condition | Percentage of household living in other type of housing not meant for inhabitation (cellar, attics) |
| Social Status | Low education population | Population with low education (no middle-school diploma) |
| | People living alone | Number of single-person households |
| Economic Status | Low-income population | Population in a household with children and no money-earning members |
| | Unemployed | Unemployment rate |


The spatially varied indicators are freely available in the census records (i.e. sub-city level) from the Italian national statistical institute (ISTAT, 2021) for three different years (1991, 2001 2011). Given the data time constraints, vulnerability is thus derived for these three years only.

The methodology to quantify vulnerability uses the equal weight analysis (EWA, e.g. Liu et al, 2020). Firstly, the individual indicators are standardized between 0 and 1, prior to aggregation (their sum); the standardization is done at the city level for the three years of record (1991, 2001, 2011) based on Equation 7:

$$\text{Standardized Indicator } (t) = \frac{\text{Indicator}(t) - \min\left(\text{Indicator}_{1991,2001,2011}\right)}{\max\left(\text{Indicator}_{1991,2001,2011}\right) - \min\left(\text{Indicator}_{1991,2001,2011}\right)}$$

(7)

Secondly, the EWA is performed according to Equation 8:

$$\text{Vulnerability } (t) = \frac{\sum \text{Standardized indicator}(t)}{\text{number of indicators}}$$

310 (8)

This approach was chosen as it is the simplest method for weighing the vulnerability indicators and it is commonly applied in the literature with regards to HWs and CWs (e.g. Buscail et al., 2012; Buzási, 2022).

Finally, we created yearly varying vulnerability maps for the period 1980-2018 following

the same linear interpolation approach used for the population.

## 3.5 Risk Quantification

Risk is a function of hazard, exposure and vulnerability, multiplied to quantify risk

(UNDRR, 2021). This is one of the two most commonly used approaches in literature

(Dong et al., 2020; Quader et al., 2017; Russo et al., 2019), with the other approach

being the addition of the different risk components. Multiplication when compared to

addition is found to better highlight the complex relationship between the different

components, due the multiplication of the multivariate probabilities of independent

variables following a product law (El-Zein and Tonmoy, 2015; Estoque et al., 2020;

Peng et al., 2017).

The risk is calculated as per Dong et al. (2020) (Equation 9):

$$\text{Risk} = \sqrt[3]{\text{Hazard} * \text{Exposure} * \text{Vulnerability}}$$

(9)

with each of the risk components having a value in [0,1]. The hazard is computed as the

probability of occurrence of HWs/CWs using the fitted Tweedie distributions probability

function for each pixel. Exposure is the standardized population density. The

vulnerability derived from standardized variables is also between [0,1]. The resulting risk

is therefore bound by 0 and 1, with 0 corresponding to the lowest level of risk and 1 to

the highest level of risk.

The risk is calculated at the municipality level because it is the lowest level of resolution

of the three elements that compose it.

In order to further investigate which are the driving factors of the risk, we disentangle the marginal effect of each component (i.e. hazard, exposure, and vulnerability) for both HWs and CWs. In turn, one of them is allowed to vary across 1980-2018 and two of them are kept constant (to their value at the year 2003, the middle of the analyzed

period).

## 3.6 Trend analysis & statistical significance

The trends are analyzed using the robust regression technique (Huber, 2011) which is often used to assess trends in natural hazards (Formetta and Feyen, 2019 for multiple hazards and Kishore et al., 2022 specifically for HWs). Robust regression seeks to

overcome part of the limitations of traditional regression analysis.

For example, the linear regression least squares method is optimal when the regression's assumptions (normal distribution, independence, equal variance) are valid (Filzmoser and Nordhausen, 2021; Khan et al., 2021). This method can be sensitive to outliers or if normality is dissatisfied (Khan et al., 2021; Brossart et al., 2011). The

robust regression method is designed to limit the effect that invalid assumptions have on the regression estimates (Filzmoser and Nordhausen, 2021; Alma, 2011).

To confirm the statistical significance of the trends, the false discovery rate (FDR) methodology is used according to Wilks (2016) and Leung et al. (2019), with a significance level $\alpha=0.05$. The FDR is defined as the statistically expected fraction of

null hypothesis test rejections at the grid cell for which the respective null hypotheses are true (Wilks 2016).

## 4 Results

### 4.1 Hazard quantification and trends

For HWs hazard intensities, the most notable year on record (1980-2018) in the region is 2003, where HWMId reached a pixel maximum of 30.4 and a median value of 16.9 over the area (Figure 2). The second most intense HW occurred in 2015 and the third most intense in 1983. Out of the six years with the highest median HWMId between 1980 and 2018, four occurred in the last decade (2010, 2013, 2015, 2017), suggesting that climate change is already increasing the frequency of heat waves in the Trentino Alto-Adige region. For CW, only 1985 stands out, with a maximum and median CWMId of 27 and 14.5, respectively, or nearly three times more than that of any other year on record. The second strongest cold wave occurred in 2012.

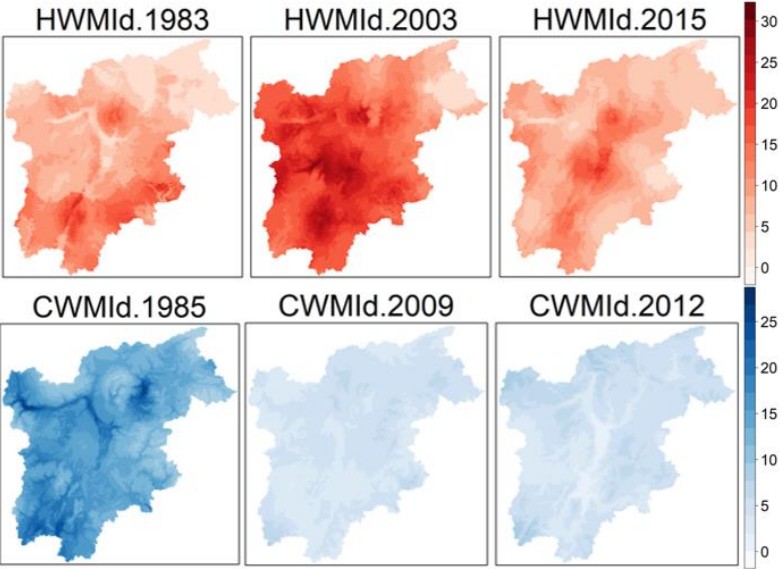

Figure 2: Regional Heat wave Magnitude Index daily (HWMId) and Cold Wave Magnitude index daily (CWMId) maps for single years with the highest regional average on record (1980-2018)

The KS tests p-values (Figure S2 in the supplementary material), indicate that the fitting of the Tweedie distribution with power parameter values between [1,2] cannot be rejected for both HWMId and CWMId. This enables us to estimate return levels for both

HWs and CWs and analyze trends based on them. The return levels for return periods of 5 years (HW5Y, CW5Y) and years (HW10Y, CW10Y) for every pixel are shown in Figure S3 in the supplementary material.

Fitting the robust linear model to the HWs values, statistically significant positive trends are found for HWs (i.e. HWMId > 0) and HWs with a magnitude larger than the 5-year

event (HWMId > HW5Y) in most pixels of the region (Figure 3). For rarer events, those larger than the 10-year event (HWMId > HW10Y), no statistically significant increase in HWs intensity are found in the region. Regarding location of these trends, some of the highest elevation parts of the region have the greatest coefficient of increase (i.e. north of Bolzano and in the mountains located in the north-west of the region). For all CWs,

we do not find statistically significant trends in any part of the region.

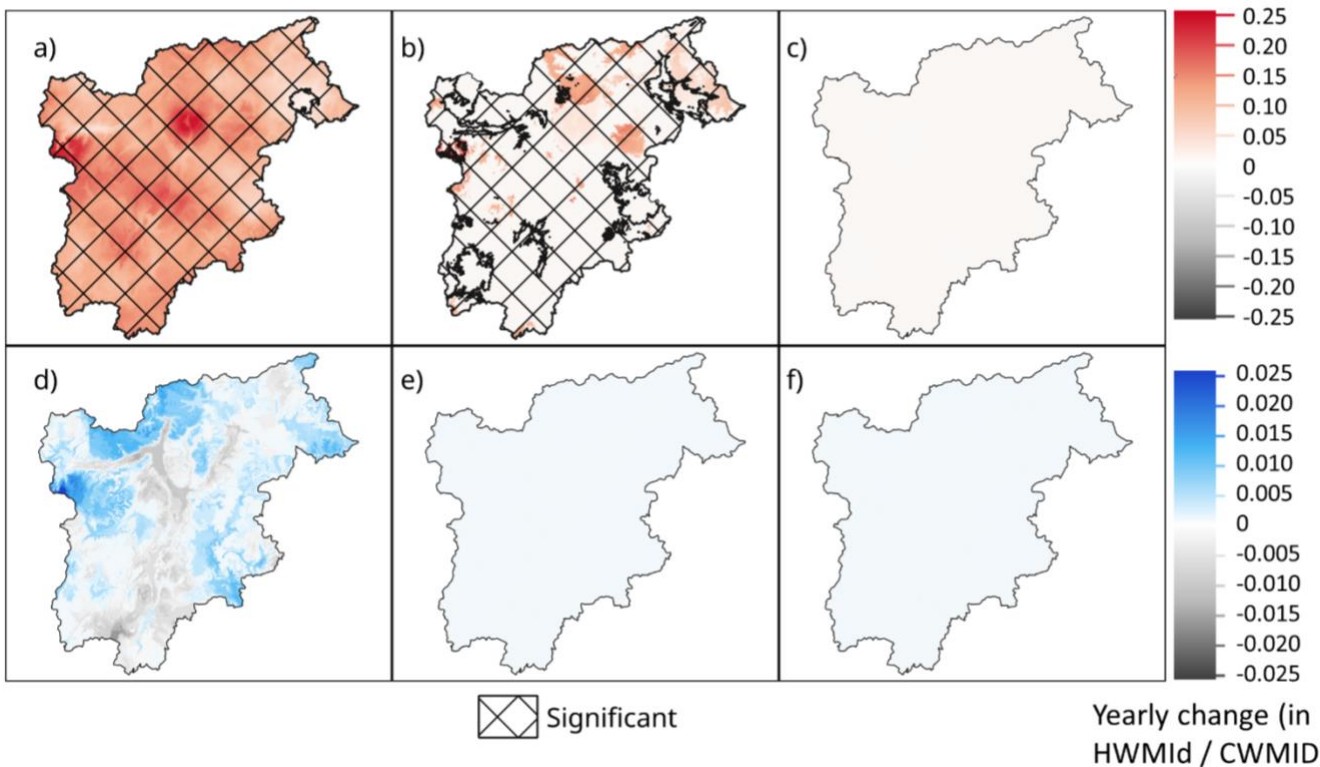

Figure *3*: Trends in heat waves (HWs) and cold waves (CWs) using the robust linear model based on yearly HWMId and CWMId magnitudes from 1980 to 2018 for HWs a) with HWMId > 0, b) with HWMId > HW5Y, c) HWMId > HW10Y and for CWs and d) with CWMId > 0, e) with CWMId > CW5Y, f) CWMId > HW10Y

## 4.2 Population exposure

Summing the overall number of people exposed over intervals (i.e. one person can be exposed each year and therefore counted multiple times over the interval), between 1980 and 2000 in the study region, about 900 000 people were exposed to a 5-year HW event, 250 000 to 10-year HW event, 3 million to 5-year CW event and 1.9 million to 10-year CW event. More recently, between 2000 and 2018, the population exposure values increased significantly to over 5 million for 5-year HW event and to about 2.5 million for

10-year HW event but the numbers decreased for CW events, to 2.4 million for 5-year

CW event and to 500 000 for 10-year CW event. Due to the importance of the

demographic change in the region over the full study period (increase of population by

23%), it is important to analyze the percentage of population impacted by these different

events. This will help us to disentangle what is driving these changes, e.g. whether

these changes are due to demographic changes or to the change in the frequency of

events, or both.

Figure 4**Error! Reference source not found.** presents the share of the population

exposed to HWs and CWs intensities larger than those of 5-year and 10-year events

over the period 1980 to 2018 on a yearly basis. It shows that a higher share of the

population was exposed to HWs more frequently after 2000 compared to the first two

decades (80s and 90s). For both return periods, the robust linear model indicates a

significant increase in the share of population exposed to HWs across the region, with a

coefficient for the increase of nearly 1% per year for HWs>HW5Y and 0.02% for

HWs>HW10Y. We did not find a significant trend in human exposure to CWs in the

region.

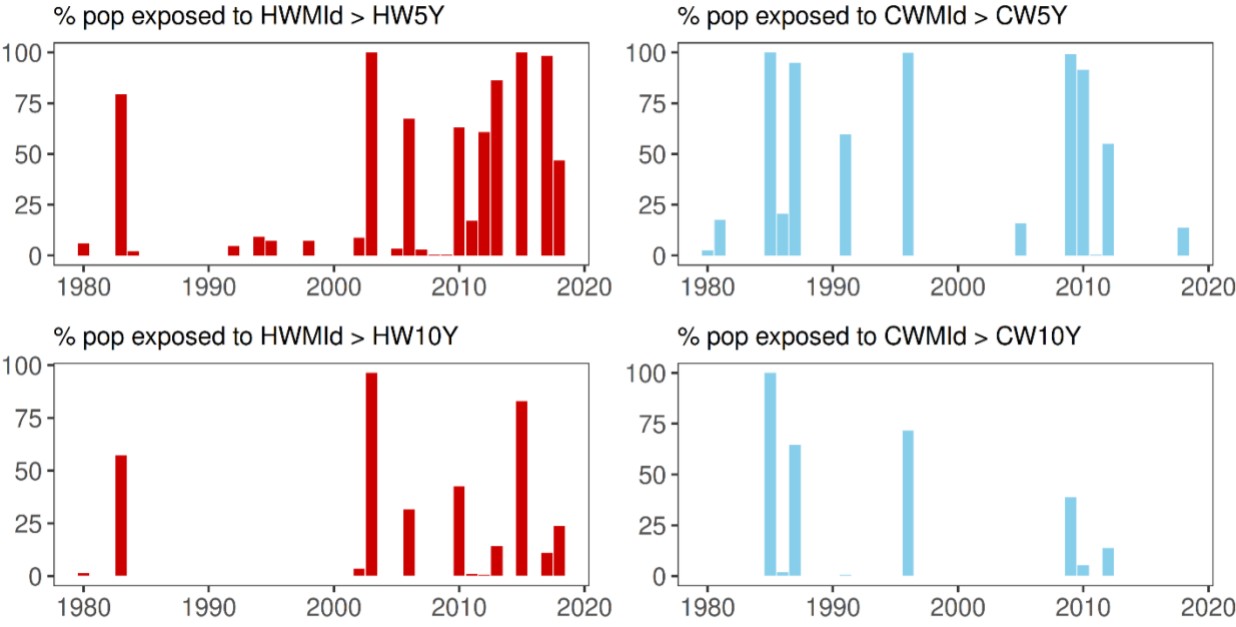

Figure 4: Percentage of population exposed to heat wave and cold wave events greater than the return levels of 5years and 10years over the span of 1980-2018

### 4.3 Vulnerability quantification

The vulnerability for the region (Figure 5) decreases with time, with an average value of 0.42 in 1991, 0.32 in 2001 and 0.27 in 2011. The main reason for the decrease in

vulnerability at regional scale is the improvement in overall education level and housing conditions (i.e., fewer people living in old and poor housing conditions). By contrast, for the larger cities (those with a population over 30,000: Merano, Bolzano, Trento, Rovereto), the vulnerability increased from 0.28 in 1991, to 0.30 in 2001, and 0.32 in 2011 (with vulnerability values averaged for those cities; see Figure S4 In the

supplemental material). The increase in these cities' vulnerability relates to the rise in age (i.e. the older age indicator) and change in social status; with time, there is a growing portion of the population above 55 and an increase in the number of people living alone in isolated households.


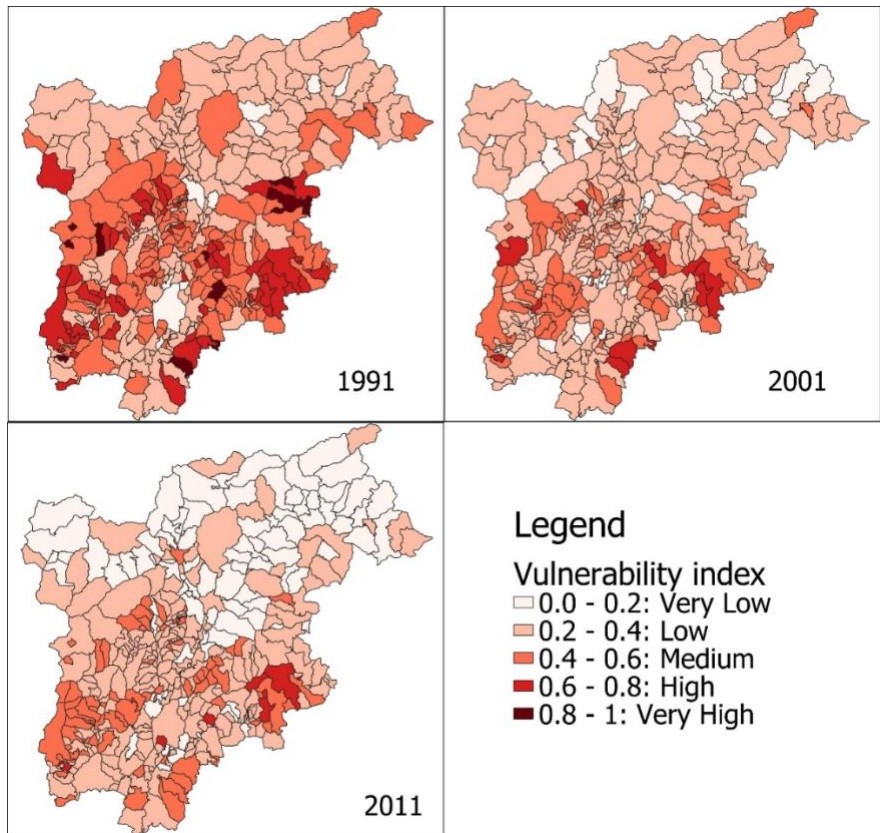

Figure 5: Calculated extreme temperatures vulnerability index for the three years of the

census records (1991, 2001, 2011) with the borders of the municipalities in black

### 4.4   Risk quantification

Figure 6 shows the trend in risk for the whole region over the period 1980-2018. The

robust linear model shows a significant increasing trend for HW risk in 40% of the

region's area, with a significant decreasing risk in some isolated parts of the region of

study. While the risk from CWs has decreased over most of the region since the 1980s,

an increase is found in the major cities (Trento, Rovereto, Bolzano and Merano).

Decadal means of the annual regional risk values confirm these trends, with the HW risk

increasing from 0.119 in the 1980s to 0.133 for the 2010s, while CW risk has decreased

from 0.134 in the 1980s to 0.124 in the 2010s. Decadal means of HW risk for the large

cities show a stronger trend compared to the whole region. We found that the average

HW risk in the main cities increased by nearly 45% compared to the 12% increase in

the whole region. Decadal means of CWs risk for the main cities increased by nearly

17% whereas in the whole region, it decreased by 7%.

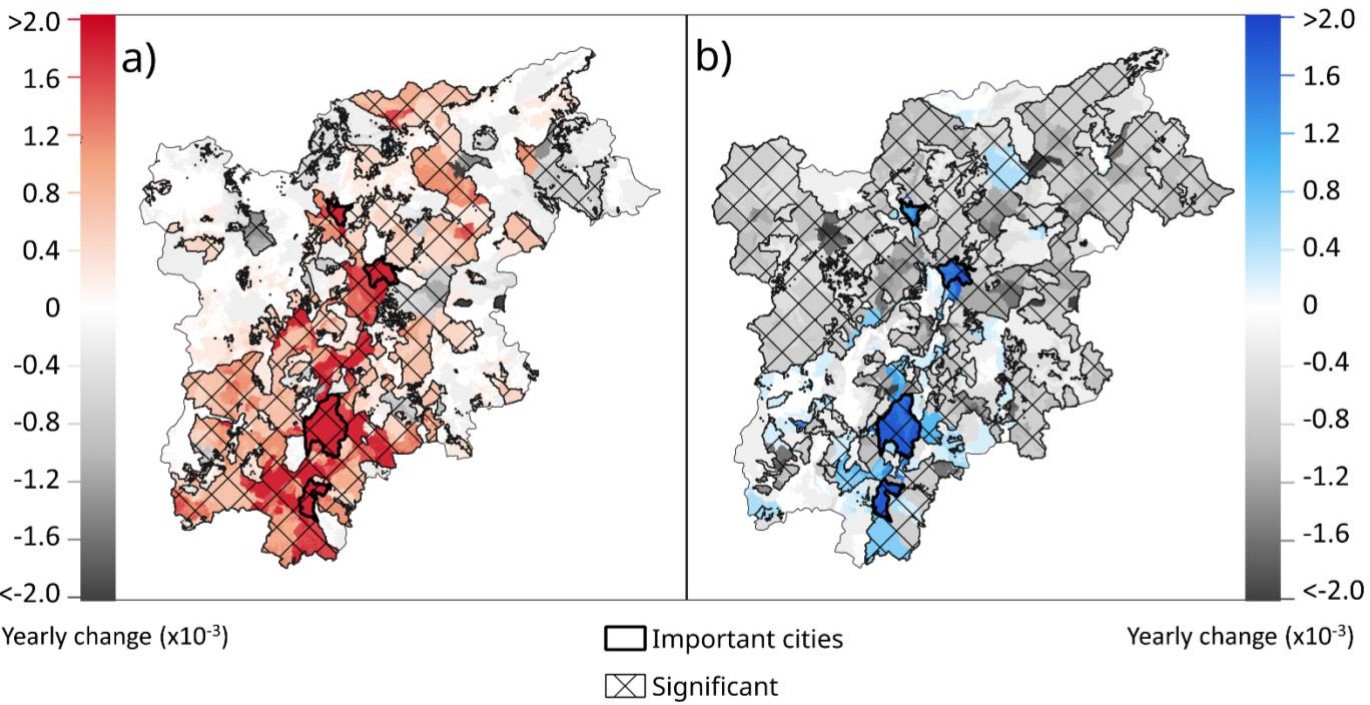

Figure 6: Trends between 1980 and 2018 of a) heat waves and b) cold waves risks

using the robust linear method, colors indicating an increase in the risk and grey a

decrease, significance is indicated with the hashing, the yearly change being the robust

linear model coefficient.

The highest annual risk levels for both HWs and CWs coincide with the years with the

highest hazard intensity (2003 for HW and 1985 for CW, see Figure S5 in the

supplementary material), indicating that the hazard is potentially the main factor for risk. However, risks are of course further modulated by exposure and vulnerability. The risks are found to be the highest in the largest cities (Bolzano, Merano, Rovereto and Trento).

Figure 7 shows the marginal effect of the driving factor behind the trends in HWs and CWs risks. Figure 7a, Figure 7c, and Figure 7e (Figure 7b, Figure 7d, and Figure 7f) show the trend in HWs (CWs) risks with only vulnerability, only exposure, and only hazard changing, respectively.

Figure 7a and Figure 7b show trends in risk due to changes in vulnerability only,

effectively indicating the locations of the increases/decreases in risk due the changes of vulnerability indicators, that are equally weighted (seen in Figure 5). These trends are found to be increasing in the main cities and nearby areas and are found to be decreasing for the rest of the region.

Figure 7c and Figure 7d show trends in risk due to change in exposure only, indicating

the locations of changing risk due to the changes in population (exposed) only. The HW and CW risks are found to be increasing in/near urban areas and decreasing in zones at high elevations and far from the urban centers.

Figure 7e shows the trends in HWs risk due to hazard only, with statistically significant increasing trends being more evident in and around highly populated areas. The figure

shows that hazard is the main driver of risk for HWs, with the significant increasing hazard trends cancelling (as can be seen in Figure 6a) most of the significant decreasing trends of the other two elements (exposure and vulnerability) seen in the Figure 7a and 7c.

Finally, Figure 7f shows no significant trends in CWs risk due to change in hazards only.

The figure indicates that the combination of three elements of the risk equation

(Equation 9) is the main driver of its risk (Figure 6b) rather than the CWs hazard only.

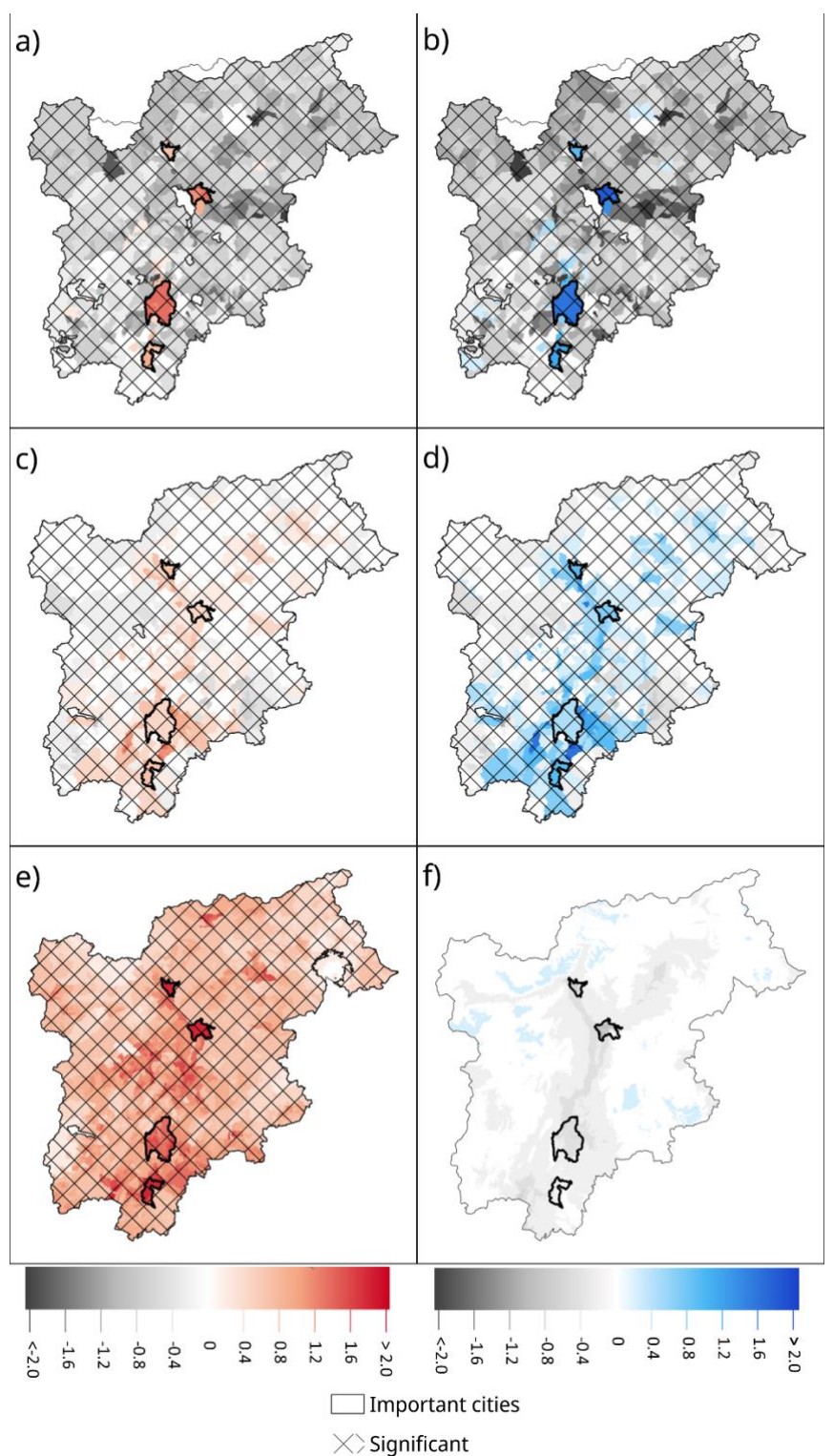

Figure 7: Trends between 1980 and 2018 of heat waves (and cold waves) risks due to
changes in: a (b) vulnerability only, c (d) exposure only, and e (f) hazard only. Trends

found with the robust linear method, colors indicating an increase in the risk and grey a

decrease, significance is indicated with the hashing, the yearly change (x10$^{-3}$) being the

robust linear model coefficient.

## 5   Discussion

The hazard analysis presented in this paper relies on the Crespi et al. (2021) air

temperature database. Although Crespi et al. (2021) is based on a state-of-the-art

interpolation approach and represents the best product for the area, more attention

should be given to measuring meteorological variables in orographically complex areas

and at high elevation. A more in-depth analysis of this sort will in turn reduce uncertainty

in spatial interpolation and improve the quantification of hazards such as HWs and CWs

and related risks.

The findings of this study agree with Russo et al. (2015), which found the greatest HWs

in the region in 1983, 2003 and 2015 in their analysis of Europe since 1950. The fact

that four of the six largest HWs occurred in the last decade suggests that climate

change is already influencing the intensity and frequency of HWs in the Trentino Alto-

Adige region. Regarding CWs, Jarzyna & Krzyżewska, (2021), also found cold spells in

the years 1985 and 2012 using different methodologies for other locations throughout

Europe. Similarly, other studies found 1985 to be a year of an exceptional CW in

Europe (Spinoni et al., 2015; Twardosz and Kossowska-Cezak, 2016).

Figure 3a indicates that a strong increase in heatwave trends is observed in the

northwest and the north of our study area. Both areas are at a high elevation (between

~1000m and ~3900m) and one includes the highest mountain in the analyzed area.

These results are consistent to those presented by Acquaotta et al., (2015), which found higher increases in temperatures at higher elevations in north-west Italy.

Our results for HWs are also in line with the finding of Bacco et al., (2021) that analyzed

trends in temperature extremes over northeastern regions of Italy (including Trentino Alto-Adige) based on homogenized data from dense station networks. Bacco et al., (2021) also found widespread warming, with significant positive trends in maximum-related mean and daytime temperature extremes. The lack of trend in CWs events is also in agreement with previous research that could not detect any trend in extreme

cold spells (Jarzyna and Krzyżewska, 2021; Piticar et al., 2018).

The trends in vulnerability and their absence of statical significance strongly depend on the available data. In our case the data used are from the  specific national census carried out every ten years and aggregated at the city spatial scale. These data are freely available and allow us  to quantify the vulnerability to natural hazards, which is a

crucial component for the risk quantification (e.g. Formetta and Feyen, 2019, Frigerio & De Amicis, 2016).

Consistently with previous studies in other European regions (e.g. (López-Bueno et al., 2021; Poumadère et al., 2005), we found that the elderly population and isolation were the indicators most affecting the increase in extreme temperature vulnerability.

The results of our vulnerability analysis contrast with the findings of Frigerio & De Amicis (2016), who report increasing vulnerability for municipalities of the Bolzano province and slightly decreasing to steady vulnerability in the Trento province. The contrast between their findings and ours, is related to the use of different indicators (i.e.

they use employment, social-economic status, family structures, race/ethnicity, and

population growth) and also a different methodology for calculating the vulnerability. The

methodology used by Frigerio & De Amicis (2016) normalize indicators across all of

Italy; by contrast in this study we normalize indicators over the Trentino Alto-Adige

region only, allowing us to better characterize local vulnerability.

Our findings on the increase in HW risks are consistent with Smid et al., (2019), which

showed an increase of risk in both current and the future period for European capitals;

the same study highlights a future decrease in CWs risk for these same cities. We found

that CWs risk is still increasing for the main cities of our study. This is also the case for

other cities in mountainous regions, such as is highlighted by López-Bueno et al. (2021)

for the metropolitan area of Madrid, where the urban area was found to be the more at

risk from CWs compared to the rural areas in the same region.

Our analysis of the risk trends shows that hazard and vulnerability are the main driving

factors of HW risk in the region of study. The changes in HW risk due to hazard also

highlight the presence of an urban heat island effect in the most populated cities of the

region (in Figure 7e these are the zones of the highest increasing trends in risk). This

has also been found in other studies of urban areas (e.g. Morabito et al., 2021). The

changes in CW risk are explained by the demographic changes (ie. an increasing and

aging population) and by other vulnerability changes, which are increasing in/around

urban areas and decreasing elsewhere.

The changes found in HW and CW risk due to changes in exposure or vulnerability only

is partially explained by rural-urban migration and by an aging population. Findings of

rural-urban migration and aging populations are presented in other studies such as

(Reynaud and Miccoli, 2018) who demonstrated these in Italy and more specifically our study area.

## 6    Summary and conclusions

Our study is one of the first to calculate risks of HWs and CWs and their trends at the community and city level for a region over a 39-year period. This is done by first quantifying the historical hazard of extreme temperature events using HWMId and CWMId indicators, at high spatial resolution (250 m) in the Trentino Alto-Adige region for the period 1980-2018. The hazard probability of occurrences is then quantified by

fitting the Tweedie distribution to HWMId and CWMId values, explicitly accounting for zero values in their time series. Two types of population exposure are found using different hazard return levels (5 years and 10 years return level). Vulnerability is calculated using 8 different socioeconomic indicators. Combining these findings, the spatio-temporal HWs / CWs risk over the time-period and at the city level is calculated.

Over the past 4 decades, HWs, i.e. HWMId>0, (and extreme HWs, i.e. HWMId>HW5Y) showed increasing trends in most of the region, with 98% (70%) being statistically significant. This results in an increasing exposure of people to extreme heat spells. For CW, we did not find a trend in hazard frequency and intensity and exposure to extreme cold remain constant. With regards to risk, a steady increase (~12%) in HWs risk and a

decrease (~7%) in CWs risk are found for the entire region. However, in larger cities of the region, a much stronger rise in HWs risk (~45%) and CWs risk (~17%) occur. This is linked with demographic changes and the social status of city inhabitants, with an increasing and ageing population living in cities and an increase in the number of one person households.

The findings of this work show that municipalities and cities in the Trentino Alto-Adige

region have experienced increasing HW risk over the timeframe 1980-2018, while

potentially experiencing a steady level of CW risk. Our detailed analysis shows where in

the region to prioritize risk mitigation measures to reduce hazard and vulnerability.

Measures to mitigate heat in cities include, for example, greening of cities (Alsaad et al.,

2022; Taleghani et al., 2019), while vulnerability could be decreased by improving the

social and living conditions of citizens, especially of the elderly who are more vulnerable

to HWs (Orlando et al., 2021; Poumadère et al., 2005; Vu et al., 2019), particularly in

the cities of this region where their share of the population is growing. If detailed data

are available for temperature, exposure and vulnerability indicators, the methodology

presented here could be applied to other regions inside and outside of Italy to help steer

local investments in climate change adaptation at the city level.

**Code availability**

The code used for calculating HWMId and CWMId is free and open source, it is the

extRemes package of R which is available here: https://cran.r-

project.org/package=extRemes.

**Data availability**

All data used in this study is available freely and openly online. The temperature

data(Crespi et al., 2021) is available at the following location:

https://doi.pangaea.de/10.1594/PANGAEA.924502. The population data from the GHSL

is available at this location: https://data.jrc.ec.europa.eu/collection/ghsl. The indicator

data used to calculate the vulnerable is available from ISTAT: https://www.istat.it/en/.

**Acknowledgments**

Giuseppe Formetta acknowledges funding from the Italian Ministry of Education, University and Research (MIUR) in the frame of the Departments of Excellence Initiative 595 2023-2027. This study was carried out also within the PNRR research activities of the consortium iNEST (Interconnected North-Est Innovation Ecosystem) funded by the European Union Next-GenerationEU (Piano Nazionale di Ripresa e Resilienza (PNRR) – Missione 4 Componente 2, Investimento 1.5 – D.D. 1058  23/06/2022, ECS_00000043). This manuscript reflects only the Authors' views and opinions.

**Author contribution**

All the authors equally contributed to the paper.

**Competing interests**

The authors have no competing interests.

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
