# Peer review of "Trends in heat and cold wave risks for the Italian Trentino Alto-Adige region from 1980 to 2018"

_Natural Hazards and Earth System Sciences, 2022_

## Referee Comment (RC1)

**General comments**

The paper presents a 38-year quantification of the risks from heatwaves (HW) and coldwaves (CW) in the region of Trentino-Alto Adige in Italy. In precise, the authors try to quantify hazard, exposure, and vulnerability from HW and CW using the Heat Wave Magnitude Index daily/Cold Wave Magnitude Index daily, the Tweedie zero-inflated distribution, high-resolution maps of the population, and a set of eight socioeconomic indicators. They claimed that this new method for the calculation of human risk from HW and CW is applicable to other regions. The manuscript has an important aspect as it offers an additional contribution to understanding the spatio-temporal risk of HW / CW. Although, I have several comments on the methodology and the presented results. In general, the level of discussion is almost minimal, while most of the statements are too often very general, missing any proper citation and profound discussion that would put their results in a comparative context. My impression is that the paper is incomplete and can be improved. I, therefore, recommend that the paper goes a major revision, and the authors need to respond to the issues I list below before the paper can be accepted for publication in NHESS

**Main comments**

1. The abstract is very extended. It must be much shorter including only the key points of the manuscript.

2. I would propose a reconstruction of the introduction. It does not have coherence, especially when going from one paragraph to another, and it is extended compared to the other sections. The novelty of the study is not being appropriately highlighted. Concerning novelty, the authors could also emphasize the advantages of applying specifically the form of Tweedie for the zero-inflated distribution. The limitations of this method should be accounted and properly included in the manuscript.

3. Lines 153-160. The used gridded temperature dataset includes uncertainties due to the interpolation of the observed data. Have the authors considered how these uncertainties may impact the results of their study?

4. In line 165, it is not clear how the cumulative indices are calculated and how someone can interpret these indices. I assume that the HWMId is the sum of the daily magnitude of the most severe heatwave in each year, something that is not clear in the manuscript.

5. Line 215. What are exactly the outcomes of the Tweedie distribution? I assume it is only the return period. Please be more clear in the manuscript.

6. Lines 213-219. These lines need more analysis as they are essential for the computation of the return period. Also, the authors must include abbreviations for the legends in the Fig S-1

7. Lines 218-219. Why have the authors chosen to keep only 5 and 10 return periods? Most extreme episodes may fall into a higher return period (e.g. 20 or 30 return years)

8. In line 253, the authors claim that the vulnerability is computed only for precise years while exposure has been calculated for each year. In line 275 the authors said that the computation of the risk was made based on the closest year. This limitation must be highlighted in the results (line 370 and further). Also, why the authors have chosen to use the "closest year" and not to interpolate the data?

9. Line 280. This section must be divided into subsections in an organized structure in order to be more clear and effective when presenting the findings of the paper. Also, the discussion section must be clearer in order to defend your research and to emphasize the significance of your research.

**Specific and minor comments:**

1. Please insert the proper citations in lines 71-72.

2. Line 76: Is there an advantage to defining hazards by return period? Please add the information at the introduction or the methodology section.

3. There is a piece of misleading information in the citations in lines 136 and 162

4. Line 178. Please revise the sentence

5. Lines 189-190. Please revise the sentence

6. Lines 235-237. Please clarify better this sentence

7. Line 255. Please elaborate on this

8. Line 269. It is not clear in the manuscript how the hazard is defined

9. Line 282. Why the authors have chosen the median and not the mean for the intensity of the HW?

10. Line 343. The authors must comment on the uncertainty in increasing and decreasing values found for vulnerability. Also, they must highlight that these trends are not statistically significant.

11. Fig 4. The vulnerability is calculated for hw or cw?

12. Line 413. "HW have occurred more frequently and have become more intense". This sentence is not properly justified in the results section.

13. Line 417. Please rephrase in order to highlight the limitations of this result

14. Line 428. Why "will be exposed"? This work is not a future projection analysis

---

## Author Comment (AC1)

The paper went through a deep revision in which we: i) revised the language, ii) modified the paper structure (separating the discussion section from the result section, highlighting the main finding in the conclusion section), iii) improved the abstract, introduction, and methodology sections, iv) we changed and improved the figures' quality and captions.

Finally, we extended the level of discussion and we added more citation to justify and compare with our findings.

Moreover, we added the new analysis requested by the reviewer n1 and n. 2, specifically:

a) As both the reviewer asked, we used the methodology of linear interpolation in time instead of the closest-year method presented in the original version of the paper

b) to disentangle the effect of each single components of the risk on its total changes, changing in turn one by one each element (i.e. vulnerability, exposure and hazard) and keeping constant the other two

c) finally we also improved our trend analysis and statistical significancy evaluation by using the FDR methodology.

Answers to the Reviewer 1

We thank the reviewer for the revision and the useful comments and insights. We reviewed the paper according to the suggestions and below you can find a one-to-one answer. The answer to the reviewer comments are provided in red, the new revised sentences are provided in blue.

General comments

The paper presents a 38-year quantification of the risks from heatwaves (HW) and coldwaves (CW) in the region of Trentino-Alto Adige in Italy. In precise, the authors try to quantify hazard, exposure, and vulnerability from HW and CW using the Heat Wave Magnitude Index daily/Cold Wave Magnitude Index daily, the Tweedie zero-inflated distribution, high-resolution maps of the population, and a set of eight socioeconomic indicators. They claimed that this new method for the calculation of human risk from HW and CW is applicable to other regions. The manuscript has an important aspect as it offers an additional contribution to understanding the spatio-temporal risk of HW / CW. Although, I have several comments on the methodology and the presented results. In general, the level of discussion is almost

minimal, while most of the statements are too often very general, missing any proper citation and profound discussion that would put their results in a comparative context. My impression is that the paper is incomplete and can be improved. I, therefore, recommend that the paper goes a major revision, and the authors need to respond to the issues I list below before the paper can be accepted for publication in NHESS

We thank the Reviewer for the valuable and constructive feedbacks, which have been very much appreciated. The paper has undergone the suggested major revision in which all the suggestions have been included. Please see below the one-to-one answers to the reviewer comments.

Main comments

1. The abstract is very extended. It must be much shorter including only the key points of the manuscript.

Thank you for your feedback on this, it has been shortened and the key points are better highlighted.

The old abstract is:

Heat waves (HW) and cold waves (CW) can have considerable impact on people. Mapping risks of extreme temperature at local scale accounting for the interactions between hazard, exposure and vulnerability remains a challenging task. In this study, we quantify human risks from HW and CW at high resolution for theTrentino-Alto Adige region of Italy from 1980 to 2018. We use the Heat Wave Magnitude Index daily (HWMId) and a Cold Wave Magnitude Index daily (CWMId) as temperature-based indicators and apply a Tweedie zero-inflated distribution to derive hazard intensities and frequencies. The hazard maps are combined with high-resolution maps of population, for which the vulnerability is quantified at community and city level using a set of eight socioeconomic indicators. We find a statistically significant increase in HW hazard and exposure, with 6.0-times more people exposed to extreme heat after 2000 compared to the last two decades of the previous century. CW hazard and

exposure remained stagnant over the studied period in the region. We observe a general trend towards increased resilience to extreme temperature spells over the region. In the larger cities of the region, however, we find that vulnerability has increased due to an ageing population and more single households. HW risk has risen practically everywhere in the region, indicating that the reduction in vulnerability in the smaller communities is outpaced by the increase in HW hazard. In the large cities, HW risk levels in the 2010s are 50% larger compared to the 1980s due to the rise in both hazard and vulnerability. Whereas in smaller communities, stagnant CW hazard and declining vulnerability results in reduced CW risk levels, the risk level in cities grew by 20% due to the increased vulnerability over the study period. The findings of our study are highly relevant for steering investments in local risk mitigation measures, while the method can be applied to other regions that have detailed information on hazard, exposure and vulnerability indicators.

The revised version abstract is below:

Heat waves (HWs) and cold waves (CWs) can have considerable impact on people. Mapping risks of extreme temperature at local scale accounting for the interactions between hazard, exposure and vulnerability remains a challenging task. In this study, we quantify risks from HWs and CWs for the Trentino-Alto Adige region of Italy from 1980 to 2018 at high spatial resolution. We use the Heat Wave Magnitude Index daily (HWMId) and the Cold Wave Magnitude Index daily (CWMId) as the hazard indicator. To obtain HWs and CWs risk maps we combined: i) occurrence probability maps of the hazard, ii) normalized population density maps, and iii) normalized vulnerability maps based on eight socioeconomic indicators. The occurrence probability of the hazard is obtained using the Tweedie zero-inflated distribution.

The methodology allowed us to disentangle the effects of each component of the risk to its total change.

We find a statistically significant increase in HWs hazard and exposure while CWs hazard remained stagnant in the analyzed area over the study period. A decrease in vulnerability to extreme temperature spells is observed trough the region except in the larger cities where vulnerability has increased. HWs risk increased in 40% of the region, with it being stronger in highly populated areas. Stagnant CWs hazard and declining vulnerability result in reduced CWs risk levels, with exception of the main cities where it grew due to their increased vulnerabilities and exposures.

The findings of our study are relevant to steer investments in local risk mitigation, and this method can potentially be applied to other regions that have similar detailed data.

2. I would propose a reconstruction of the introduction. It does not have coherence, especially when going from one paragraph to another, and it is extended compared to the other sections. The novelty of the study is not being appropriately highlighted. Concerning novelty, the authors could also emphasize the advantages of applying specifically the form of Tweedie for the zero-inflated distribution. The limitations of this method should be accounted and properly included in the manuscript.

Thank you for your suggestion on this, your constructive feedback has been taken into account, the introduction has been shortened and has been rearranged with a better emphasis on the point mentioned. The Introduction of the revised paper is structured as follows:
1) importance of HWs and CWs from a global to the local scale
2) definition of the HWs and CWs risk as product of hazard, exposure, and vulnerability
3) how the single risk components have been computed in different studies and what are the main challenges in defining them as well introducing Tweedie as a possible solution to one of these challenges, i.e., accounting for zero inflation.
4) The need to move to a high-resolution risk analysis and goals and objectives of our study

The advantages and limitations of using a tweedie methodology has been highlighted as well and is present both in the introduction as well as in the new limitation section of this study. This has been done with the following new sentences:

The main advantage of the Tweedie distribution is the possibility of considering many distributions for the continuous and semi-continuous domain such as: normal, Gamma, Poisson, Compound Gamma-Poisson, and Inverse Gaussian (Bonat and Kokonendji, 2017; Rahma and Kokonendji, 2021; Shono, 2008; Temple, 2018). Moreover, for some of these distributions (i.e. Poisson mixtures of gamma distributions) it explicitly enables the fitting of zero-inflated data. Tweedie distribution main limitation is the complex distribution's fitting methodology and the difficulties to compare it to other models via information criteria such as the Akaike's information criterion (Shono, 2008)

3. Lines 153-160. The used gridded temperature dataset includes uncertainties due to the interpolation of the observed data. Have the authors considered how these uncertainties may impact the results of their study?

We thank the editor for the comment. We added into the revised paper a more detailed description of the interpolation methods that Crespi et al. (2021) used and a quantification of the errors they obtained in a leave one out cross validation framework. The new sentence is reported here:

"This dataset is based on more than 200 station daily records which have been quality controlled and homogenized. The interpolation method is based on a combination of 30-year temperature climatology (1981–2010), daily anomalies and explicitly accounts for topographic features (i.e. elevation, slope) which are crucial in orographic complex areas such as the Trentino Alto-Adige. The leave one out cross validation presented in Crespi et al. (2021) finds mean correlation coefficient higher than 0.8 and mean absolute errors of around 1.5 degree Celsius (on average across months and stations used for the interpolation)."

Although the dataset is based on a state-of-the-art approach and the errors found in cross validation as relatively small, we added a new sentence in the conclusion section of the paper

where we underline the importance of reducing the uncertainty in interpolating temperature data in orographically complex area. The new sentence is reported here:

"The hazard analysis presented in this paper rely on the Crespi et al. (2021) air temperature database. Although it is based on a state-of-the-art interpolation approach and it represents the best product for the area, more attention should be given to measuring meteorological variables in orographically complex area and at high elevation. This will in turn reduce the uncertainty in spatial interpolation and improve the quantification of impacting hazards such as HWs and CWs."

4. In line 165, it is not clear how the cumulative indices are calculated and how someone can interpret these indices. I assume that the HWMId is the sum of the daily magnitude of the most severe heatwave in each year, something that is not clear in the manuscript.

We thank the reviewer for the comment. We have re-organized the section of the hazard definition according to the reviewer suggestions. Moreover, we also gave a practical example to explain what the index value means. The new sentence is:

"To quantify the hazard we used the HWMId (Russo et al., 2015) and the CWMId (Smid et al., 2019). These indices represent a way of measuring extreme temperature events while considering their durations, intensity, and taking in account the site-specific historical climatology (30years).
According to Russo et al. (2015), HWMId is defined as the maximum magnitude of the HWs in a year. A HW occurs when the air temperature is above a daily threshold for more than three consecutive days. The threshold is set to the 90$^{th}$ percentile of the temperature data of the day and the window of 15 days before and after throughout the reference period 1981-2010. The magnitude of a HW is the sum of the daily heat magnitude $HM_d$ of all the consecutive days composing the HW (Equation 1):

$$HM_d(T_d) = \begin{cases} \dfrac{T_d - T_{30y25p}}{T_{30y75p} - T_{30y25p}} & \text{if } T_d > T_{30y25p} \\ 0 & \text{if } T_d \leq T_{30y25p} \end{cases}$$

(1)

where HM$_d$(T$_d$) corresponds to the daily heat magnitude, T$_d$ the temperature of the day in question and T$_{30y25p}$ and T$_{30y75p}$ correspond to the 25$^{th}$ and 75$^{th}$ percentile of the yearly maximum temperature for the 30 years of the reference period (1981-2010). The interquartile range (IQR, i.e. the difference between the T$_{30y75p}$ and T$_{30y25p}$ percentiles of the daily temperature) is used as the heatwave magnitude unit and represents a non-parametric measure of the variability of the temperature timeseries. Therefore, a value of $HM_d$ equals to 3 means that the temperature anomaly on day d with respect to T$_{30y25p}$ is 3 times the IQR. Finally, for a given year HWMId corresponds to the highest sum of magnitude (HMd) over the consecutive days composing a heatwave event (with only days with HMd > 0 considered). Analogously to the HWMId, CWMId is defined as the minimum magnitude of the CWs in a year (Smid et al., 2019). A CW occurs when the air temperature is below a daily threshold for more than three consecutive days. The threshold is set to the 10th percentile of the temperature data of the day and the window of 15 days before and after throughout the reference period 1981-2010.

The daily cold magnitude corresponds to (Equation 2):

$$CM_d(T_d) = \begin{cases} \dfrac{T_d - T_{30y75p}}{T_{30y75p} - T_{30y25p}} & \text{if } T_d < T_{30y75p} \\ 0 & \text{if } T_d > T_{30y75p} \end{cases}$$

(2)

where CM$_d$(T$_d$) corresponds to the cold daily magnitude, T$_d$ the daily temperature and T$_{30y25p}$ and T$_{30y75p}$ correspond to the 25$^{th}$ and 75$^{th}$ percentile yearly temperature for the 30 years used as a reference. Inversely to HWMId, the lowest cumulative magnitude sum is retained for each year and with only consecutive days with CM$_d$ < 0 considered to calculate it. CWMId being always <T 0, its absolute values are retained for its values to be on a positive interval (similar to HWMId)."

5. Line 215. What are exactly the outcomes of the Tweedie distribution? I assume it is only the return period. Please be more clear in the manuscript.

We thank the reviewer for the question. We add a new section where we specify the outcomes of the Tweedie distribution and the functions we used in the paper. This is connected to the next comment (6). The new sentence is:

6. Lines 213-219. These lines need more analysis as they are essential for the computation of the return period. Also, the authors must include abbreviations for the legends in the Fig S-1

We thank you for the suggestion. We added the following new section to specify better how we fitted the Tweedie distribution and performed parameter estimation:

"It provides distribution density, distribution function, quantile function, random generation for the Tweedie distributions. The Tweedie parameters (i.e. mean, power, and dispersion) have been estimated by the "tweedie.profile" function (Dunn, 2015) using the maximum likelihood as described by Dunn (2015) and Dunn and Smyth (2005)."

Further we modified the figure S1, shortening the legends as requested by the reviewer and being more explicit in the caption to clarify the abbreviations used in the figure.
Old figure and caption:

[Figure]

Figure S - 1: Cumulative distribution functions for both HWMId / CWMId at the location of the cities of Bolzano and Trento

The new figure and the new caption is reported below:

[Figure]

Figure S2: Cumulative distribution functions (CDF) for both HWMId / CWMId at the location of the cities of Bolzano and Trento, displaying the probability (P) showing the empirical cumulative distributions (ECDF) for these locations as well as the confidence interval (CI) of the power value of the Tweedie distribution.

7. Lines 218-219. Why have the authors chosen to keep only 5 and 10 return periods? Most extreme episodes may fall into a higher return period (e.g. 20 or 30 return years)

We thank the reviewer for the comment. We choose 5 and 10 years return period for accounting of both the length of the analyzed return period (39 years) and the type of hazards we are analyzing (the HWs and CWs usually doesn't occur every year). Higher return periods estimations could be affected by higher extrapolation effects and more uncertainty.
We add the following sentence in the paper to clarify this point. The new sentence is:

"This choice aims to account for of both the length of the analyzed period (39 years) and the type of hazards we are analyzing (HWs and CWs usually doesn't occur every year). Higher return level estimations would be affected by extrapolation effects and higher uncertainty."

8. In line 253, the authors claim that the vulnerability is computed only for precise years while exposure has been calculated for each year. In line 275 the authors said that the computation of the risk was made based on the closest year. This limitation must be highlighted in the results (line 370 and further). Also, why the authors have chosen to use the "closest year" and not to interpolate the data?

We thank the reviewer for this question. We intensively revised this part of the paper. To account for the reviewer suggestion, we interpolated the data in time and removed the approximation of using the closest year (when possible) for all the variables (i.e. hazard, vulnerability and exposure).

The exposure data (i.e. population) are available for the years 1975, 1990, 2000 and 2015. We created yearly varying population maps following the methodology presented in other studies (e.g. Formetta and Feyen, 2019; Neumayer and Barthel, 2011). We linearly interpolated the data in time for the period 1980 to 2015 (assuming a constant rate in between available years) and we used the closest year for the period 2016-2018.

The vulnerability data are available for the years 1991, 2001, 2011. We created yearly varying vulnerability maps following the same approach we used for the population: we interpolated the data in time for the period 1991-2011 (assuming a constant rate in between available years) and we used the closest year for the period 1980-1990 and 2012-2018.

We added the following sentence in the section of the exposure:
"To more accurately model exposure, we created yearly varying population maps for the period 1980-2018 following the methodology presented in other studies (e.g. Formetta and Feyen, 2019; Neumayer and Barthel, 2011). We linearly interpolated the data in time for the period 1980 to 2015 (assuming a constant rate in between available years) and we used the closest year for the period 2016-2018."

We added the following sentence in the section of the vulnerability:

"Finally, we created yearly varying vulnerability maps for the period 1980-2018 following the same approach we used for the population."

9. Line 280. This section must be divided into subsections in an organized structure in order to be more clear and effective when presenting the findings of the paper. Also, the discussion section must be clearer in order to defend your research and to emphasize the significance of your research.

We thank you very much for this constructive feedback. To properly respond to this suggestion, we have re arranged the structure of the paper. We subdivided the results section in four specific subsections: 4.1) Hazard quantification and trends 4.2) Population exposure 4.3) Vulnerability quantification and 4.4) Risk quantification.

Finally, the discussion section has been separated from the result section and deeply reviewed to better emphasize as per the reviewer's comment. The new discussion section is:

[revised manuscript text omitted]

**Specific and minor comments:**

1. Please insert the proper citations in lines 71-72.

Thank you for the comment. We added the new references. The old sentence in the paper was: "Most of these studies have found increasing trends in exposure to HW and for the studies that also analyzed CW, found decreasing trends for them."

The revised sentence reads:

"These studies found increasing trends in HWs (Chambers, 2020; Dosio et al., 2018) and decreasing trends in CWs in their period of analysis (Oldenborgh et al., 2019, Smid et al., 2019)."

2. Line 76: Is there an advantage to defining hazards by return period? Please add the information at the introduction or the methodology section.

We thank the reviewer for the question. We used the return period because it is a standard way to express extreme events intensity. The main novelty is that for the first time, we used the Tweedie zero inflated distribution to quantify the cumulative distribution function of the HWMId and CWMId, which are indeed zero inflated data.

3. There is a piece of misleading information in the citations in lines 136 and 162

We thank the reviewer for the comment. The sentence has been modified according to the reviewer's suggestion and is now consistent:

[revised manuscript text omitted]

9. Line 282. Why have the authors chosen the median and not the mean for the intensity of the HW?

The median was chosen to avoid the possibility of a particular high or low intensity area affecting the overall result.

10. Line 343. The authors must comment on the uncertainty in increasing and decreasing values found for vulnerability. Also, they must highlight that these trends are not statistically significant.

We thank the reviewer for the suggestion we added the following sentence to the discussion section:

The trends in vulnerability and their absence of statical significance strongly depend on the available data. In our case they are the output of specific national census carried out every ten years and aggregated at the city spatial scale. From the other side, these data represent a freely available option to quantify the vulnerability to natural hazards, which is a crucial component for the risk quantification (e.g. Formetta and Feyen, 2019, Frigerio & De Amicis, 2016).

11. Fig 4. The vulnerability is calculated for hw or cw?

The vulnerability is calculated for extreme temperatures so both hw and cw Several other studies used the same approach, see for example the methodology used in Nwoko (2016) and Török et al. (2021).

Nwoko, D. S. V. I. for E. T. R. in N.: Developing social vulnerability index for newcastle extreme temperatures, Msc Thesis, Durham University, 68 pp., 2016.

Török, I., Croitoru, A.-E., and Man, T.-C.: Assessing the Impact of Extreme Temperature Conditions on Social Vulnerability, Sustainability, 13, 8510, https://doi.org/10.3390/su13158510, 2021.

12. Line 413. "HW have occurred more frequently and have become more intense". This sentence is not properly justified in the results section.

We agree with the reviewer comment, and we rephrased the sentence.

The sentence old sentence was: HW have occurred more frequently and have become more intense.

The new sentence is: "HWs, i.e. HWMId>0, (and extreme HWs, i.e. HWMId>HW5Y) showed increasing trends in most of the region, with 98% (70%) being statistically significant."

13. Line 417. Please rephrase in order to highlight the limitations of this result

We thank the reviewer for the comment. In the new revised paper this part have been moved in the discussion section and it has been rephrased according to this comment.

The old sentence was:

"In general, vulnerability is decreasing over time in the Trentino Alto-Adige region. However, in the larger cities of the region, vulnerability is increasing due to an ageing population and more single households. It should be noted that the socioeconomic indicators of vulnerability are only available for three points in time, which does not allow to do a proper trend analysis of vulnerability"

The new sentence in the discussion section is:

"The trends in vulnerability and their absence of statical significance strongly depend on the available data. In our case they are the output of specific national census carried out every

ten years and aggregated at the city spatial scale. From the other side, these data represent a freely available option to quantify the vulnerability to natural hazards, which is a crucial component for the risk quantification (e.g. Formetta and Feyen, 2019, Frigerio & De Amicis, 2016)."

14. Line 428. Why "will be exposed"? This work is not a future projection analysis

We revised the sentence according to the reviewer suggestion. The old sentence was:

"The findings of this work shows that municipalities and cities in the Trentino Alto-Adige region, but likely also in many other regions, will be exposed especially to more frequent and intense heat, while potentially still experiencing the same levels of cold wave hazard"

The revised sentence is:

"The findings of this work shows that municipalities and cities in the Trentino Alto-Adige region have been seen increasing trends in HWs risk over the timeframe 1980-2018, while potentially experiencing the same levels of CWs risk."

---

## Author Comment (AC2)

Reply to the commentary:

Dear Authors,

After reading your paper, we think that the discussion of your results could benefit from the findings shown in our work "Recent changes in temperature extremes across the north-eastern region of Italy and their relationship with large-scale circulation. Climate Research, 81, 167-185" (Di Bacco and Scorzini (2020); doi: 10.3354/cr01614), in which we analyzed trends in temperature extremes over northeastern regions of Italy (including Trentino Alto-Adige) based on homogenized data from dense station networks.

Anna Rita Scorzini and Mario Di Bacco

We thank the Professors Anna Rita Scorzini and Mario Di Bacco for the suggestion. We added the following sentence in the discussion part of our revised paper to account for their new findings in line with ours. The new sentence is:

"Our results for HWs are also in line with the finding of Bacco et al., (2021) that analyzed trends in temperature extremes over northeastern regions of Italy (including Trentino Alto-Adige) based on homogenized data from dense station networks. They also found widespread warming, with significant positive trends in maximum-related mean and daytime temperature extremes".

---

## Author Comment (AC3)

The paper went through a deep revision in which we: i) revised the language, ii) modified the paper structure (separating the discussion section from the result section, highlighting the main finding in the conclusion section), iii) improved the abstract, introduction, and methodology sections, iv) we changed and improved the figures' quality and captions.

Finally, we extended the level of discussion and we added more citation to justify and compare with our findings.

Moreover, we added the new analysis requested by the reviewer n1 and n. 2, specifically:

a)  As both the reviewer asked, we used the methodology of linear interpolation in time instead of the closest-year method presented in the original version of the paper
b)  to disentangle the effect of each single components of the risk on its total changes, changing in turn one by one each element (i.e. vulnerability, exposure and hazard) and keeping constant the other two
c)  finally we also improved our trend analysis and statistical significancy evaluation by using the FDR methodology.

Answers to the Reviewer 2

General comments
This study investigates and quantifies hazard, exposure, and vulnerability to heat and cold extremes in the Italian region Trentino Alto-Adige for 1980-2018 and calculates the resulting combined risk. The structure of the paper is generally clear, and the presented results are mostly convincing. My main comments concern 1) the language, 2) a more precise estimation of the contribution of hazard, exposure, and vulnerability to the overall risk, 3) extending the figure captions, and 4) adjusting the p-values of the statistical significance tests to control for the false discovery rate.

We thank the reviewer for the revision and the useful comments and insights. We reviewed the paper according to the suggestions and below you can find a one-to-one answer. The answer to the reviewer comments are provided in red, the new revised sentences are provided in blue.

Main comments:

Although the manuscript is generally well comprehensible, the structure of some sentences and some of the terms that are used make some parts difficult to understand. I would thus recommend to carefully check the whole text again with a special focus on rephrasing cumbersome sentences (some examples are listed under "specific comments")

We revised the manuscript and its organization as well as the results presentation, discussion, and conclusions. All the specific comments have been implemented in the revised paper.

I think that it would be possible to calculate the contribution of changes in hazard, exposure, and vulnerability to the overall changes in risk ratio (e.g. by keeping exposure constant while changing the other parameters, and similarly for the other parameters). I think this could provide a valuable insight into the importance of climate change vs population and socioeconomic changes.

We implemented the analysis requested by the reviewer, we provided a new figure summarizing the result of this analysis and we added the following new sentences in the revised paper.

The new figure is the following:

[Figure]

Figure 6: Trends of heat waves (and cold waves) risks due to changes in: a (b) vulnerability only, c (d) exposure only, and e (f) hazard only. Trends found with the robust linear method, colors indicating an increase in the risk and grey a decrease, significance is indicated with the hashing, the yearly change being the robust linear model coefficient.

The new sentences added in the result section are the following:

"Figure 6 shows the marginal effect of the driving factor behind the trends in HWs and CWs risks. Figure 6-a, Figure 6-c, and Figure 6-e (Figure 6-b, Figure 6-d, and Figure 6-f) show the trend in HWs (CWs) risks with only vulnerability, only exposure, and only hazard changing, respectively.

The results in Figure 6-a and Figure 6-b show the same patterns as well as Figure 6-c and Figure 6-d because exposure and vulnerability are the same for both HWs and CWs and hazard is the only differing variable.

Figure 6-a (Figure 6-b) show increasing trends in risk (due to change in vulnerability only) in the main cities and nearby areas. Decreasing trends are found for most of the remaining region.

Figure 6-c (Figure 6-d) show increasing trends in risk (due to change in exposure only) in/near urban areas and decreasing trends in zones at high elevations and far from the urban centers.

Figure 6-d show that the hazard is the main driver of risk for HWs, with statistically significant increasing trends, more evident in and around highly populated areas. Finally, Figure 6-e show no significant trends in CWs risk (due to change in hazards only)."

The new sentences added in the discussion section are the following:

The analysis of the trends of risk while changing only one of its three variables and keeping constant the remaining two shows that hazard and vulnerability are the main driving factor of the HWs risk. The changes in HWs risk due to hazard also highlights the presence of urban heat island in the most populated cities of the region (in Figure 6-e these are the zones of the highest increasing trends in risk). This has also been found in other in urban areas (e.g. Morabito et al., 2021). The changes in CWs risk is mainly explained by the demographic and vulnerability changes, which are increasing in/around urban areas and decreasing elsewhere. The changes found in HWs and CWs risk due to changes in exposure or vulnerability only is partially explained by rural-urban migration and an aging population, which is presented in other studies such as (Reynaud and Miccoli, 2018).

The captions of the figures are currently very short and contain insufficient information to fully understand the associated figures. A caption should be written such that it is possible to understand a figure and its main message only from watching the figure and reading its caption (i.e., without the need to read the main text). I would thus recommend extending the captions such that they explain the figures and the displayed features more comprehensibly.

Thank you, this has been considered and the captions have been rewritten accordingly. Please, see also the specific comments where we show the modification.

Many of the figures contain estimates of statistical significance. As the multiple statistical tests (which I presume are conducted independently for each grid cell) may cause to overestimate the statistical significance (Wilks, 2016, https://doi.org/10.1175/BAMS-D-15-00267.1), I would suggest adjusting the p-values by controlling for the false discovery rate as proposed by Wilks (2016).

Thank you for this suggestion, all the figures have been remade with this suggestion, their significance corresponds to the FDR significance. Moreover, we added the following new section in which we explained the methodology we used.

The new sentence is:

The trends are analyzed using the robust regression technique (Huber, 2011). This method is often used throughout the literature for natural hazards (Formetta and Feyen, 2019; Kishore et al., 2022).

The trends are analyzed using the robust regression technique (Huber, 2011). This method is often used throughout the literature for assessing trends in natural hazards (Formetta and Feyen, 2019 for multiple hazards and Kishore et al., 2022 specifically for HWs). To confirm the statistical significance of the trends the false discovery rate (FDR) methodology is used according to Wilks (2016) and Leung et al. (2019), with a significance level $\alpha=0.05$. The FDR is defined as the statistically expected fraction of null hypothesis test rejections at the grid cell for which the respective null hypotheses are actually true (Wilks 2016).

Specific comments:

1  Lines 15-16 (and generally for the description of the Tweedie distribution): I think it would make sense to first mention that HWMId and CMWId are normalized to the interval (0, 1)

to combine them with the exposure and vulnerability metrics, and only then write that the Tweedie distribution is used for this purpose.

Thank you this has been adjusted. The old sentence was:

We use the Heat Wave Magnitude Index daily (HWMId) and a Cold Wave Magnitude Index daily (CWMId) as temperature-based indicators and apply a Tweedie zero-inflated distribution to derive hazard intensities and frequencies. The hazard maps are combined with high-resolution maps of population, for which the vulnerability is quantified at community and city level using a set of eight socioeconomic indicators.

The new sentence is:

To obtain HWs and CWs risk maps we combined: i) occurrence probability maps of the hazard, ii) normalized population density maps, and iii) normalized vulnerability maps based on eight socioeconomic indicators. The occurrence probability of the hazard is obtained using the Tweedie zero-inflated distribution. The methodology allowed us to disentangle the effects of each component of the risk to its total change.

2   Line 17: Maybe better "which are used to derive vulnerability"

Thank you for this suggestion, this sentence has been rephrased entirely as part of the changing of the abstract. See the sentence above.

 3  Line 18 ff: I am wondering how the increased resilience is determined? Maybe the factors causing the increased resilience could be mentioned here (same for CW)

We thank the reviewer for the suggestion, the sentence was actually removed in order to make the abstract shorter per the other reviewer's request, only the trends in vulnerability are now mentioned.

The old sentence was: "We observe a general trend towards increased resilience to extreme temperature spells over the region. In the larger cities of the region, however, we find that vulnerability has increased due to an ageing population and more single households."

The new revised sentence is:

"A decrease in vulnerability to extreme temperature spells is observed trough the region except in the larger cities where vulnerability has increased."

4   Line 36 (and other occasions): I think that the text would be easier to read if an "s" would be added to the acronyms for the plural forms of "heat wave" and "cold wave" (i.e., HWs, CWs).

We thank the reviewer for the suggestion, we adjusted it throughout the entire text.

5   Line 38: In which direction do they change? Increasing or decreasing?

We thank the reviewer for the question. We revised the section. The old section "With global warming, heat and cold wave intensities and durations are expected to change (Perkins-Kirkpatrick and Gibson, 2017; Russo et al., 2015; Smid et al., 2019), which could increase the risks to society."

The new revised sentence is:

"With global warming, HWs intensities and durations are expected to increase while those of CWs are expected to decrease (Perkins-Kirkpatrick and Gibson, 2017; Russo et al., 2015; Smid et al., 2019), which would change their risks to society."

6   Line 42: How are heatwaves defined in this study? Based on percentiles? Or is it HWMId?

We thank the reviewer for the question. Heatwave in that study are defined as 3days above the $90^{th}$ percentile temperature. This is now mentioned in our article and the new sentence is: "In Europe, recent high intensity HWs events (2003 and 2018, where HWs are defined as 3 days over 90th temperature percentile of the 1980-2010)."

7   Line 43-44: This part of the sentence about GCP losses is a bit difficult to understand. I would suggest rephrasing it.

The sentence about the GDP has been removed in our attempt to make the introduction a bit shorter and straightforward as suggested by both reviewers.

8   Line 71-73: Rephrase, as the last part reads rather cumbersome.

We thank the reviewer for the suggestion. The sentence has been rephrased. The old sentence was:

"Most of these studies have found increasing trends in exposure to HW and for the studies that also analyzed CW, found decreasing trends for them."

The new sentence is:

"These studies found increasing trends in HWs (Chambers, 2020; Dosio et al., 2018) and decreasing trends in CWs in their period of analysis (Oldenborgh et al., 2019, Smid et al., 2019)."

References were added as per the other reviewer's request.

9   Line 99: Maybe "are most exposed to" instead of "affect"

We thank the reviewer for the question, we revised the sentence. The old sentence was:
"In Korea at the county level, Kim et al. (2017) found that elderly living alone, agricultural workers and unemployed affect vulnerability to heat wave days and tropical nights"
The new revised sentence is:
"In Korea at the county level, Kim et al. (2017) found that elderly living alone, agricultural workers, and unemployed are the most significant vulnerability factors to extreme temperatures."

10 Line 113-114: What does "normalized population" mean? Can this be shortly explained here?

We thank the reviewer for the question, and we revised the sentence. Russo et al., 2019 normalized the population density maps in order to have values between 0 and 1 and therefore consistent with the hazard (between 0 and 1) and the vulnerability (between 0 and 1) in the risk equation.
The old sentence was: "where the exposure is the normalized population".
The new sentence is: "where the exposure is the population density normalized in [0;1] based on its maximum, minimum values;"

11 Line 134: Remove "for the"
We thank the reviewer for the suggestion, and we removed it.
The old sentence was:
"The aim of this article is to solve some of these previous limitations while quantifying heat and cold waves hazards, the human exposure, vulnerability, and risk for the at the high-definition city scale for the Trentino-Alto-Adige region over the period 1980-2018"
The new revised sentence is:

"The aim of this article is to solve some of these previous limitations while quantifying HW and CW hazards, the human exposures, vulnerabilities, and risks at the high-definition (i.e. city-scale) over the period 1980-2018, for the Trentino-Alto-Adige region"

12 Lines 141-143: Something with the reference to Figure 1 is wrong

We thank the reviewer for the comment. We have addressed revising the sentence. The old sentence was:

"The Trentino Alto-Adige region (**Error! Reference source not found.**) is a mountainous region in northern Italy, which borders Austria"

The new revised sentence is:

"The Trentino Alto-Adige region (Figure 2) is a mountainous region in northern Italy, which borders Austria"

13 Lines 145-146: I think it would be good to exactly state the population of Trento, Bolzano, Merano and Rovereto

We thank the reviewer for the suggestion. We revised the sentence accordingly. The old sentence was: "Its most populous cities are the two provincial capitals -Trento and Bolzano - as well as minor cities Merano and Rovereto (both have a population of over 30000)"

The new revised sentence is

Its most populous cities (population for 2022 in parenthesis) are the two provincial capitals, Trento (118509) and Bolzano (107025), as well as minor cities such as Merano (40994) and Rovereto (39819).

14 Lines 157-160: I think it would be good to shortly explain which variables are used for the extrapolation of the temperature dataset (e.g. height, land cover, something else?)

We thank the reviewer for the question. We revised this section including more information on the interpolation schema (and on the geomorphological variables used in the interpolation).

The old sentence was:

"The dataset is obtained with the anomaly-based approach taking into account elevation of the local station observations; the dataset has undergone a quality analysis and control against the stations' observations (Crespi et al. 2021)."

The new revised sentence is:

"This dataset is based on more than 200 station daily records which have been quality controlled and homogenized. The interpolation method is based on a combination of 30-year temperature climatology (1981–2010), daily anomalies and explicitly accounts for topographic features (i.e. elevation, slope) which are crucial in orographic complex areas such as the Trentino Alto-Adige. The leave one out cross validation presented in Crespi et al. (2021) finds mean correlation coefficient higher than 0.8 and mean absolute errors of around 1.5 degree Celsius (on average across months and stations used for the interpolation)."

15 Lines 164ff: What is the reference period for calculating HWMId? I would also explicitly mention that data are pooled from a window of 15 days before and after each day (currently this is not entirely clear).

We thank the reviewer for the question. We revised the sentence better specifying the reference period for calculating HWMId. The old sentence was:

For HWMId, from the temperature time series in each grid cell, we select the days where the temperature is above the 90th percentile of the dataset Ad (Equation 1):

$$A_d = \bigcup_{y=1981}^{2010} \bigcup_{i=d-15}^{d+15} T_{y,i}$$

(1)

where y corresponds to the year, i to the day, and Ty,i correspond to the temperature of the corresponding year and day and the dataset Ad corresponds to the temperature data for 30 years, centered on a 31-day window for the day in question. Three consecutive days above this threshold correspond to a HW.

The new revised sentence is:
According to Russo et al. (2015), HWMId is defined as the maximum magnitude of the HWs in a year. A HW occurs when the air temperature is above a daily threshold for more than three consecutive days. The threshold is set to the 90$^{th}$ percentile of the temperature data of the day and the window of 15 days before and after throughout the reference period 1981-2010.

16 Line 175: I think rather "daily heat magnitude"

We thank the reviewer for the suggestion. We revised the sentence accordingly.

The old sentence was: "to the heat daily magnitude."

The new revised sentence is: "the daily heat magnitude."

17 Line 176: Are the percentiles calculated from the temperature distribution or from the yearly maximum temperatures? (the latter is done in the original publication by Russo et al.).

We thank the reviewer for the question. We used the yearly maximum temperatures, and we revised the sentence accordingly.

The old sentence was:

"where HMd(Td) corresponds to the heat daily magnitude, Td the temperature of the day in question and T30y25p and T30y75p correspond to the 25$^{th}$ and 75th percentile temperature for the 30 years used as a reference"

The new revised sentence is:

where $HM_d(T_d)$ corresponds to the daily heat magnitude, $T_d$ the temperature of the day in question and $T_{30y25p}$ and $T_{30y75p}$ correspond to the 25$^{th}$ and 75$^{th}$ percentile of the yearly maximum temperature for the 30 years of the reference period (1981-2010).

18 Line 178: I would write "only consecutive days with HMd above 0"

We thank the reviewer for the suggestion. We modified the sentence (and the entire paragraph to describe the HWMId in a more clear way). The old sentence was:

"The highest cumulative magnitude is retained for each year and only consecutive days above 0 are considered when calculating it".

The new revised sentence is:

Finally, for a given year HWMId corresponds to the highest sum of magnitude (HMd) over the consecutive days composing a heatwave event (with only days with HMd > 0 considered).

19 Line 189-190: I think it would be good to explicitly write that based on the definition used in this paper, CMd is always <0

We thank the reviewer for the question. We revised the sentence according to the suggestion.

The old sentence was:

"Similarly, the lowest cumulative magnitude is retained for each year and only consecutive days below 0 are considered when calculating it. For both the values of HWMId and CWMId

to be positive and on the same interval, the absolute values of CWMId are retained from this point on."

The new revised sentence is:

Inversely to HWMId, the lowest cumulative magnitude sum is retained for each year and with only consecutive days with $CM_d < 0$ considered to calculate it. CWMId being always <T 0, its absolute values are retained for its values to be on a positive interval (similar to HWMId).

20 Lines 210-212: This is partly a repetition, maybe shorten it?

We thank the reviewer for the suggestion, we have removed the sentence accordingly.

Line 220ff: I would suggest writing more specific what the KS test has been used for in this paper ("statistical fit verification" sounds rather generic)

We thank the reviewer for the suggestion. We modified the sentence accordingly.

Old sentence: "For statistical fit verification, the Kolmogorov–Smirnov (KS) test on two samples is used with one sample being the found HWMId or CWMId values, and the other sample being a randomly generated sample using the fitted distribution value."

New revised sentence.

"The goodness of fit of the Tweedie distribution fitted to the HWMId/CWMId data for every pixel have been tested by means of a Kolmogorov-Smirnov test of hypothesis. The test is performed using two samples, with the first being the data and the other being a randomly generated sample using the fitted distribution parameters.

21 Line 230: "population data"

We thank the reviewer for the suggestion. We modified the sentence accordingly.

Old sentence: "To quantify the population exposed to HW and CW we use time-varying population  from the Global Human Settlement Layer (GHSL) (Schiavina et al., 2019). The data is available at a resolution of 250m for the following years: 1975, 1990, 2000 and 2015:"

New revised sentence: "To quantify the population data exposed to HWs and CWs, we use time-varying population data  from the Global Human Settlement Layer (GHSL) (Schiavina et al., 2019). The population data is available at a resolution of 250m for the following years: 1975, 1990, 2000 and 2015".

22 Lines 254-256: This sentence is not clear to me. Could it be explained a bit more in detail how this was done and why this approach was chosen?

We clarified better this concept in the revised paper. Equation 8 and 9 have been added as well as paragraph mentioning why this methodology was picked.

The old sentence was: "The methodology to quantify vulnerability uses the equal weight analysis (EWA) with the indicators being standardized between 0 and 1 prior to aggregation according to Liu et al, (2020)."

The new revised sentence is:

The methodology to quantify vulnerability uses the equal weight analysis (EWA, e.g. Liu et al, 2020). Firstly, the individual indicators are standardized between 0 and 1, prior to aggregation (their sum); the standardization is done at the city level for all the years of record (1991, 2001, 2011) based on Equation 7:

$$\text{Standardized Indicator } (t) = \frac{\text{Indicator}(t) - \min(\text{Indicator}_{1991,2001,2011})}{\max(\text{Indicator}_{1991,2001,2011}) - \min(\text{Indicator}_{1991,2001,2011})}$$

(7)

Secondly, the EWA is performed according to Equation 8:

$$\text{Vulnerability } (t) = \frac{\sum \text{Standardized indicator}(t)}{\text{number of indicators}}$$

(8)

This approach was chosen as it is the simplest method for weighing the vulnerability indicators and it is commonly applied in the literature with regards to HWs and CWs (e.g. Buscail et al., 2012; Buzási, 2022).

Finally, we created yearly varying vulnerability maps for the period 1980-2018 following the same approach we used for the population.

23 Lines 274-279: Another approach could be the temporal linear interpolation of the exposure and vulnerability variables.

We thank the reviewer for this question. We intensively revised this part of the paper. To account for the reviewer suggestion, we interpolated the data in time and removed the approximation of using the closest year (when possible) for all the variables (i.e. hazard, vulnerability and exposure).

The exposure data (i.e. population) are available for the years 1975, 1990, 2000 and 2015. We created yearly varying population maps following the methodology presented in other studies (e.g. Formetta and Feyen, 2019; Neumayer and Barthel, 2011). We linearly interpolated the data in time for the period 1980 to 2015 (assuming a constant rate in between available years) and we used the closest year for the period 2016-2018.

The vulnerability data are available for the years 1991, 2001, 2011. We created yearly varying vulnerability maps following the same approach we used for the population: we interpolated the data in time for the period 1991-2011 (assuming a constant rate in between available years) and we used the closest year for the period 1980-1990 and 2012-2018.

We added the following sentence in the section of the exposure:
"To more accurately model exposure, we created yearly varying population maps for the period 1980-2018 following the methodology presented in other studies (e.g. Formetta and Feyen, 2019; Neumayer and Barthel, 2011). We linearly interpolated the data in time for the period 1980 to 2015 (assuming a constant rate in between available years) and we used the closest year for the period 2016-2018.."
We added the following sentence in the section of the vulnerability:
"Finally, we created yearly varying vulnerability maps for the period 1980-2018 following the same approach we used for the population."

24 Line 303: Here, does HW mean the yearly HWMId values or something else? Could that be specified?
We thank the reviewer for the suggestion. Yes it is correct and we modified the sentence accordingly. Old sentence:" statistically significant positive trends are found for HW in most pixels of the region (Figure 2)"
New revised sentence: "Fitting the robust linear model to the HWs values, statistically significant positive trends are found for HWs (i.e. HWMId > 0) and HWs with a magnitude larger than the 5-year event (HWMId > HW5Y) in most pixels of the region (Figure 2)."

25 Line 305-306: If I understand correctly, there should only be 3-4 values for HW10Y in each pixel, given that a period of 39 years is used. I am not sure whether a trend can be deduced from such few data points.

We agree with the reviewer comment. We used the robust regression method and the FDR method to evaluate the trend in a more robust way. For very exteme heatwave hazard in the revised paper we obtain as result no statistical significance (with FDR).

26 Line 312: Maybe "that was" instead of "and"?

Thank you, this sentence has been removed in the revised paper.

27 Lines 324-328: I would add "event" after HW and CW.

Thank you, we modified accordingly. However this section is now in the discussion section. The old sentence was: "The significant increasing trend for HW that we find are consistent with literature that reported increasing HW trends in Europe over the last decades (Perkins-Kirkpatrick and Lewis, 2020; Piticar et al., 2018; Serrano-Notivoli et al., 2022; Spinoni et al., 2015 Zhang et al., 2020). The lack of trend in CW is also in agreement with previous research that could not detect any trend in extreme cold spells (Jarzyna and Krzyżewska, 2021; Piticar et al., 2018)"

The new revised sentence is: "The significant increasing trend we found in HWs events are consistent with other studies in Europe over the last decades (e.g. Perkins-Kirkpatrick and Lewis, 2020; Piticar et al., 2018; Serrano-Notivoli et al., 2022; Spinoni et al., 2015; Zhang et al., 2020). The location of our highest increasing trends in HWs events are concordant to those of the higher increase in temperatures found at higher elevations by Acquaotta et al., (2015) in north-west Italy. Our results for HWs are also in line with the finding of Bacco et al., (2021) that analyzed trends in temperature extremes over northeastern regions of Italy (including Trentino Alto-Adige) based on homogenized data from dense station networks. They also found widespread warming, with significant positive trends in maximum-related mean and daytime temperature extremes. The lack of trend in CWs events is also in agreement with previous research that could not detect any trend in extreme cold spells (Jarzyna and Krzyżewska, 2021; Piticar et al., 2018)."

28 Lines 329-331: But Figure 3 does not present a separation of both effects! It shows the combined effects of changes in HWs and of changes in population. I think that for disentangling both effects, one of them would need to be kept constant (see also main comment above)

We implemented this change in the revised paper. See the main comment above to view the new figure and new sentences added in result and discussion sections.

29 Line 350: Not sure that "extreme age" is the right term.

We thank the reviewer for the suggestion, we modified accordingly. Old sentence:

"The increase in these cities' vulnerability relates to the extreme age indicator and social status,"

New revised sentence: "The increase in these cities' vulnerability relates to the older age indicator and social status"

30 Line 360: I would delete "somehow"

We thank the reviewer for the suggestion, and we modified that sentence accordingly from:

The results of our vulnerability analysis somehow contrast with the findings of Frigerio & De Amicis (2016), who report increasing vulnerabilities for municipalities of the Bolzano province and slightly decreasing to steady vulnerabilities in the Trento province.

The new sentence is: The results of our vulnerability analysis somehow contrast with the findings of Frigerio & De Amicis (2016), who report increasing vulnerabilities for municipalities of the Bolzano province and slightly decreasing to steady vulnerabilities in the Trento province.

31 Lines 362-365: Does this refer to the study by Frigerio & De Amicis?

We thank the reviewer for the comment, and have clarified this aspect in a clearer way. This refers to the difference between the two (our study and theirs) and has been specified. This part is now in the discussion. The old sentence was:

Old sentence was:

"The results of our vulnerability analysis somehow contrast with the findings of Frigerio & De Amicis (2016), who report increasing vulnerabilities for municipalities of the Bolzano province and slightly decreasing to steady vulnerabilities in the Trento province. This likely relates to the use of different indicators (employment, social-economic status, family structures, race/ethnicity, and population growth) and a different methodology for calculating the vulnerability. Notably in Frigerio & De Amicis (2016) the normalization of indicators is applied across all of Italy as opposed to only over the Trentino Alto-Adige region in this study, which may better characterize local vulnerability."

The new revised sentence is:

"The results of our vulnerability analysis contrast with the findings of Frigerio & De Amicis (2016), who report increasing vulnerabilities for municipalities of the Bolzano province and slightly decreasing to steady vulnerabilities in the Trento province. This contrast, between our finding and theirs, is related to the use of different indicators (employment, social-economic status, family structures, race/ethnicity, and population growth) and a different methodology for calculating the vulnerability where the normalization of indicators is applied across all of Italy in their study, as opposed to only over the Trentino Alto-Adige region in this study, the latter characterizing better local vulnerability. The selection of different indicators and methodology might yield different results."

32 Lines 368-372: These results cannot easily be seen in the figures. I would suggest to change the figures to make this better visible (see my comments to Figure 5 below)

We thank the reviewer for the comment and have remade the figure in order to make the results more evident. The difference is shown in the comment about the figure below.

33 Lines 377-380: What are the main factors? Can they be identified, and can their contribution be quantified? (see also my main comment above)

Following your main comment a further analysis has been conducted and the results are discussed in the appropriate section and are visible in Figure 6. See also the reply to the main comment above

34 Line 407: Mainly "normalized" instead of "sized"

We thank the reviewer for the suggestion. We use hazard quantification. The new sentence is:

"The hazard probability of occurrences are then quantified by fitting a Tweedie distribution to the HWMId and CWMId values, explicitly accounting for zero values in their time series"

35 Line 409: I do not really understand the meaning of this sentence.

We thank the reviewer for the comment and have revised the sentence accordingly.

Old sentence: "Exposure is be found using the different fitted hazard levels."

New sentence: "Two types of population exposure are found using the different hazard levels (5 years and 10 years return level)."

36 Line 428: Are there any proofs/studies showing that this is the case "likely also in many other regions"? Otherwise this statement should be deleted.

We thank the reviewer for the comment, and we deleted the sentence accordingly.

Figures:

1. Figure 1: I think it would be good to have some more information in the caption, e.g. that Merano, Bolzano, Trento, and Rovereto are the main cities in the region and what the colors mean.

*We thank the reviewer for the comment we added more information as quested. The old caption was:* Figure 1: The Trentino Alto-Adige region

*The new caption is:* Figure 2: The Trentino Alto-Adige region and its most populated cities (Trento, Bolzano, Rovereto and Merano); the colors indicating the elevations, river network, and lakes.

"

2. Figure S-1: The abbreviations used in the legends and the titles should be explained in the caption.

Thank you, we modified the figure and the caption accordingly.

[Figure]

Figure S - 1: Cumulative distribution functions for both HWMId / CWMId at the location of the cities of Bolzano and Trento

[Figure]

Figure S2: Cumulative distribution functions (CDF) for both HWMId / CWMId at the location of the cities of Bolzano and Trento, displaying the probability (P) showing the empirical cumulative distributions (ECDF) for these locations as well as the confidence interval (CI) of the power value of the Tweedie distribution.

3. Table 1: What are the exact definitions of "population living in at risk housing" and "population with low income"? Could you be more specific? And does the diploma/degree for "population with low education" refer to school or university degrees?

Thank you for the comment, we added more specific information in the table. Old table:

Table 1: Vulnerability indicators used (after Ho et al., 2018)

| Category | Indicator | Definition |
|---|---|---|
| Extreme Age | Older Age | Population over 55 years old |
| | Infants | Population under 5 years old |
| Household physical characteristics | People in old houses | Number of household living in housing built prior to 1960 |

| | People in poor living condition | Population living in at risk housing |
|---|---|---|
| Social Status | Low education population | Population with low education (no diploma or degree) |
| | People living alone | Number of single-person households |
| Economic Status | Low-income population | Population with low income |
| | Unemployed | Unemployment rate |

New table:

Table 2: Vulnerability indicators used (after Ho et al., 2018)

| Category | Indicator | Definition |
|---|---|---|
| Extreme Age | Older Age | Population over 55 years old |
| | Infants | Population under 5 years old |
| Household physical characteristics | People in old houses | Percentage of household living in housing built prior to 1960 (corresponding to when better insulation started being implemented) |
| | People in poor living condition | Percentage of household living in other type of housing not meant for inhabitation (cellar, attics) |
| Social Status | Low education population | Population with low education (no middle-school diploma) |
| | People living alone | Number of single-person households |
| Economic Status | Low-income population | Population in a household with children and no money-earning members |
| | Unemployed | Unemployment rate |

4. Why has the year 1960 been used for the category "people in old houses"? Is this an arbitrary choice or are there reasons to choose this year?

The year is not arbitrary, 1960 is first of all used in the study on which the indicators are based from (Ho et al, 2018). Second, the implementation of insulation dates from the 1960s, this has been specified in several studies about building insulations in several locations such as Austria or Italy (Mukati, 2021; De Angelis et al., 2020) and the first building energy regulation in Italy is from 1973 (Carrosio, 2015; Magnani et al., 2020), therefore it can be assumed that some building in an alpine region in Italy bordering Austria had insulations.

Carrosio, G.: Politiche e campi organizzativi della riqualificazione energetica degli edifici, Sociol. URBANA E RURALE, https://doi.org/10.3280/SUR2015-106002, 2015.

De Angelis, A., Ascione, F., De Masi, R. F., Pecce, M. R., and Vanoli, G. P.: A Novel Contribution for Resilient Buildings. Theoretical Fragility Curves: Interaction between Energy and Structural Behavior for Reinforced Concrete Buildings, Buildings, 10, 194, https://doi.org/10.3390/buildings10110194, 2020.

Magnani, N., Carrosio, G., and Osti, G.: Energy retrofitting of urban buildings: A socio-spatial analysis of three mid-sized Italian cities, Energy Policy, 139, 111341, https://doi.org/10.1016/j.enpol.2020.111341, 2020.

Mukati, A.: Effect of Heatwaves on the Cooling Demand of Austrian Residential Buildings, 2021.

5. Figure S-3: I wonder whether it would make sense to add the borders of the municipalities/districts shown in Figure 4 also to this figure. Moreover, I would suggest to use a linear color scale between 0.05 and 1 and add more ticks to the colormap (not just 0, 0.05, and 1)

The figure was remade to account for the reviewer comment. The old figure was:

[Figure]

HWMId Tweedie fitting KS test P-values     CWMId Tweedie fitting KS test P-values

the new revised figure is.

[Figure]

6. Figure 4: I think it would be good to add a figure (or a subplot) that shows the evolution for the four cities as it is impossible to identify them and to see their changes just from the maps. Also, what do the black borders in the figures depict? Is it municipalities or districts? This should be added in the caption.

We thank the reviewer for the comment we created a new figure and placed in the supplementary material according to the suggestion. The new added figure is:

[Figure]

Figure S5: Evolution of the vulnerabilities of the 4 large cities of the region (Merano, Bolzano, Trento and Rovereto).

Moreover, the black borders are municipality, and this was added in the caption of the revised figure. The old caption was Figure 3: Calculated vulnerability index for the three years of the census records (1991, 2001, 2011)

The new caption is:

Figure 4: Calculated extreme temperatures vulnerability index for the three years of the census records (1991, 2001, 2011) with the borders of the municipalities in black.

7. Figure 5: The trends are difficult to see due to the many hatching lines. I would suggest having only fewer hatching lines (like in Figure 2). Also, how is it possible to see that a trend is positive or negative? (the tau values are positive both for HWs and CWs). Like for Figure 4, I'd suggest adding a separate panel showing the results for the four cities.

The Figure was remade, the cities were added directly on it and are visible. The old figure was:

[Figure]

The new figure is:

---

## Referee Report (RR1)

**Reviewer1**

I would like to thank the authors for addressing all of my comments and technical corrections. The manuscript has been improved a lot after the revision process, therefore I suggest accepting the paper for publication as it is.

---

## Author Response (AR2)

The authors revised the manuscript according to several of my suggestions. The content of the manuscript is generally interesting, and I think it is worth to be published. Yet, I still have some concerns about ambiguities of several statements due to unclear language. Additionally, I would like to know why the authors chose to replace the original trend estimation by the robust linear model, as the replacement seems to change parts of the result in a non-negligible way.

The authors thank the reviewer for the time spent on a thorough review and have provided answers to the comments below.

General & minor comments:

• The results displayed in Figure 2 changed considerably compared to the first manuscript version. I guess this is due to the usage of the robust linear model instead of Mann-Kendall's test. Yet, I did not find any explanation in the author's response why this new method has been chosen (apologies in case I missed it) and why it yields quite different trend estimates. This makes me wonder a bit about the robustness of the estimated trends. Additionally, as mentioned also further below, I think it would be necessary to shortly explain in the manuscript what the robust linear model is and why it has been chosen.

Thank you for this comment.

The robust linear model (RLM) test was chosen in place of the Mann-Kendall (MK) test because it is a reliable methodology (also with regards to outliers) and it allows us to be quantitative with regards to the trends (i.e. know by how much the decrease, increase is, e.g. Huber, 2011). Moreover, we found the following other papers which also adopted the same method for heatwaves trend analysis (e.g. Kishore et al., 2022)

The trends that we found do not differ significantly because of the change in the trend test methodology (i.e. RLM or MK). The differences are due to the use of the false detection rate (FDR). We implemented FDR in the revised version of the paper following the suggestion of the reviewers, in contrast with the single fixed threshold as done in the original version of the paper.

To illustrate this, a comparison of the MK test and the RLM with respect to the same significance level assessment based on the FDR has been performed. For each pixel and for the different return levels threshold, we computed and reported in Table 1 the percentage of the study area in which the two methods provide: i) positive statistically-significant agreement; ii) negative statistically-significant agreement; iii) positive statistically non-significant agreement; iv) negative statistically non-significant agreement; v) disagreement in statistically-significant cells; vi) disagreement in statistically non-significant cells.

The highest percentage of the study area in disagreement for any of the return levels is less than 5%, for CWMId>CW5Y. The highest percentage of the study area in disagreement for statistically significant regions is less than 4%, for HWMId>HW5Y. The NA represents the cases in which both methods had no value for the corresponding row of the table. Results show that the two methods (MK and RLM) provide consistent outcomes when both are assessed using the FDR methodology.

Table 1: the percentage of the total study area in which the two methods (MK and RLM) agree or disagree.

| | HWMId > 0 | HWMId > HW5Y | HWMId > HW10Y | CWMId > 0 | CWMId > CW5Y | CWMId > CW10Y |
|---|---|---|---|---|---|---|
| i) Agreement significant positive | 98.86 | 86.87 | NA | NA | NA | NA |
| ii) Agreement significant negative | NA | NA | NA | NA | NA | NA |
| iii) Agreement non-significant positive | 0.76 | 9.42 | 91.45 | 59.34 | 75.73 | 47.46 |
| iv) Agreement non-significant negative | NA | NA | 7.7 | 39.03 | 19.89 | 52.51 |
| v) Disagreement significant perc | 0.38 | 3.71 | NA | NA | NA | NA |
| vi) Disagreement non-significant perc | NA | NA | 0.85 | 1.63 | 4.38 | 0.03 |

An explanation of the robust linear model is provided here and in the relevant minor comment below. This new sentence has been added to the paper.

Old sentence:

The trends are analyzed using the robust regression technique (Huber, 2011). This method is often used throughout the literature for assessing trends in natural hazards (Formetta and Feyen, 2019 for multiple hazards and Kishore et al., 2022 specifically for HWs).

New sentence:

The trends are analyzed using the robust regression technique (Huber, 2011) which is often used to assess trends in natural hazards (Formetta and Feyen, 2019 for

multiple hazards and Kishore et al., 2022 specifically for HWs). Robust regression seeks to overcome part of the limitations of traditional regression analysis.

For example, the linear regression least squares method is optimal when the regression's assumptions (normal distribution, independence, equal variance) are valid (Filzmoser and Nordhausen, 2021; Khan et al., 2021). This method can be sensitive to outliers or if normality is dissatisfied (Khan et al., 2021; Brossart et al., 2011). The robust regression method is designed to limit the effect that invalid assumptions have on the regression estimates (see Filzmoser and Nordhausen, 2021 and Alma, 2011 for more details).

References:

Hipel, K.W. and McLeod, A.I., (2005). Time Series Modelling of Water Resources and Environmental Systems. Electronic reprint of our book orginally published in 1994

Huber, P. J.: Robust Statistics, in: International Encyclopedia of Statistical Science, edited by: Lovric, M., Springer, Berlin, Heidelberg, 1248–1251, https://doi.org/10.1007/978-3-642-04898-2_594, 2011

• The format of references to figures is broken in several instances – please check! I encourage the authors to check the manuscript for formatting issues before (re-submission).

Thank you for this comment, this is a Mircrosoft Word issue when going from track changes to untracked changes, the authors apologies for this occurring and have resolved this in the submission for this revision.

• The language should be checked again. Particularly, the introduction was partly hard to read and understand. Sometimes, the language obscures the meaning of statements (e.g., what does "would change their risks to society" in line 34 mean? I think it would need to be "change the risks they pose to society") I anticipate that some of this will be tackled by the journal proof reading, but nevertheless it should be guaranteed that ambiguities stemming from unclear language are minimized.

Thank you for this comment.

For the sentence line 34, we agree with the comment. The authors changed this specific part of the sentence, to simplify it from:

" which would change their risks to society"

To:

"changing the risks they pose to society"

The authors have also been through the text again and have attempted to minimize the number of ambiguities. This can be viewed from the tracked changed version of the manuscript.

• Results: The results section begins with the description of several Supplementary figures, which I do not find so ideal. I think it is best practice to start the results by describing figures that are shown in the main manuscript. I am not familiar with the NHESS manuscript requirements, but I think it would be good if one or two figures could be moved from the supplementary to the main manuscript, if possible.

Thank you for this comment. We agree for the first figure that is therefore moved from the supplementary material to the main text. The presentation and the comment of this figure was already present in the main text. However, the second figure (the

new S2) that we mentioned in the result section has been kept in the in supplementary material. This figure only shows that that the KS test indicates no rejection for the entire region and is therefore not interesting to show in the main text.

• Lines 447-463: I think that Figure 6 is really an interesting figure, but it could be highlighted and explained better. The current manuscript just goes through the single figure panels one by one, without making connections between them. I think that restructuring the text by focusing on the main features of the figure and on what the message of this figure should be, would make the description much more interesting.

Thank you for this comment, the authors have restructured the text by focusing on the main features of this figure and its message. The text was changed from to

From:

Figure 6-a (Figure 6-b) show increasing trends in risk (due to change in vulnerability only) in the main cities and nearby areas. Decreasing trends are found for most of the remaining region.

Figure 6-c (Figure 6-d) show increasing trends in risk (due to change in exposure only) in/near urban areas and decreasing trends in zones at high elevations and far from the urban centers.

Figure 6-d show the hazard is the main driver of risk for HWs, with statistically significant increasing trends, more evident in and around highly populated areas. Finally, Figure 6-e show no significant trends in CWs risk (due to change in hazards only).

To:

Figure 7a and Figure 7b show trends in risk due to changes in vulnerability only, effectively indicating the locations of the increases/decreases in risk due the changes of vulnerability indicators, that are equally weighted (seen in Figure 5). These trends are found to be increasing in the main cities and nearby areas and are found to be decreasing for the rest of the region.

Figure 7c and Figure 7d show trends in risk due to change in exposure only, indicating the locations of changing risk due to the changes in population (exposed) only. The HW and CW risks are found to be increasing in/near urban areas and decreasing in zones at high elevations and far from the urban centers.

Figure 7e shows the trends in HWs risk due to hazard only, with statistically significant increasing trends being more evident in and around highly populated areas. The figure shows that hazard is the main driver of risk for HWs, with the significant increasing hazard trends cancelling (as can be seen in Figure 6a) most of the significant decreasing trends of the other two elements (exposure and vulnerability) seen in the Figure 7a and 7c.

Finally, Figure 7f shows no significant trends in CWs risk due to change in hazards only. The figure indicates that the combination of three elements of the risk equation (Equation 9) is the main driver of its risk (Figure 6b) rather than the CWs hazard only.

Specific comments:

• Line 17-18: I am not sure that the sentence about the Tweedie distribution is helpful in the abstract if the purpose of the application of the Tweedie distribution is not clearly specified. I would suggest to either add that information or remove this sentence.

Thank you for this comment, the reason for using the Tweedie distribution has been specified. The authors find it is important to mention the Tweedie distribution in the abstract given it is one of the main novelties of this paper.

The paragraph has therefore been adjusted from:

To obtain HWs and CWs risk maps we combined: i) occurrence probability maps of the hazard, ii) normalized population density maps, and iii) normalized vulnerability maps based on eight socioeconomic indicators. The occurrence probability of the hazard is obtained using the Tweedie zero-inflated distribution.

To:

To obtain HWs and CWs risk maps we combined: i) occurrence probability maps of the hazard obtained using the zero-inflated Tweedie distribution (accounting directly for the absence of events for certain years) ii) normalized population density maps, and iii) normalized vulnerability maps based on eight socioeconomic indicators.

• Line 19: I think rather "contributions" than "effects"

Thank you, this has been adjusted. The sentence went from:

The methodology allowed us to disentangle the effects of each component of the risk to its total change.

To:

The methodology allowed us to disentangle the contributions of each component of the risk to its total change.

• Line 48: Which increase in mortality does this refer to? The previous sentences do not mention any increase in mortality due to CWs.

Thank you for this comment, you are right this sentence is not needed, it was meant to be interpreted with regards to the previous sentence. This has been adjusted from:

With regards to CWs in Europe, recent winters have claimed 790 deaths in 2006 and 549 deaths in 2012 (Kron et al., 2019). The increase in mortality and among elder people is also found in Italy for CWs. For example, de'Donato et al., (2013) reported a notable increase in mortality (47%) for the timeframe of the 2012 CW in the city of Bolzano.

to:

With regards to CWs in Europe, recent winters have claimed lives with 790 deaths in 2006 and 549 deaths in 2012 (Kron et al., 2019).In Italy, de'Donato et al., (2013) report an increase in mortality (47%) for the timeframe of the 2012 CW in the city of Bolzano compared to the 4 previous winters (2008-2011).

• Lines 56-59: Might be worth to also mention what hazard refers to.

Thank you for this comment. This new sentence has been added to the main text:

Hazard is defined as a process, phenomenon or human activity that may cause loss of life, injury or other health impacts, property damage, social and economic disruption or environmental degradation and hazards being characterized by location, intensity or magnitude, frequency, and probability.

• Line 68: What does "qualitative" mean in this context? Does it mean arbitrary thresholds?

Thank you for this comment. We mean that the limits were subjective. The authors of HWs and CWs studies often time use a threshold for hazard, ie ( 0<HWMId<3, 3<HWMId<6, 6<HWMId<9), these thresholds (3, 6, 9) are subjective.

This has been specified in the sentence, changing it from:

Most studies on HWs and CWs have used qualitative numerical thresholds on the indicator to define severity and exposure to the hazards.

to:

Studies on HWs and CWs typically have used subjective numerical thresholds, on the indicator to define severity and exposure to the hazards (e.g. 0<HWMId<3, 3<HWMId<6, 6<HWMId<9).

• Lines 77-90: The description of the Tweedie distribution is rather technical. That is fine, but I think it would help the readers (at least, it would help me) if one or two sentences could be added explaining in a few simple words, why the Tweedie distribution is the right choice here (e.g., what are its main advantages for the application here?). Also, it would be nice if the implications of the limitations of the Tweedie distribution for this study (lines 88-90) could be briefly explained.

Thank you for this comment. The authors have made clearer why the Tweedie distribution is the right choice, as well as specified the implications of the limitations.

The paragraph has changed from:

Instead, for the first time we use the zero-inflated distribution of Tweedie families (Jorgensen, 1987; Tweedie, 1984) to estimate HWs and CWs frequency of occurrence, which enabled us to directly account for the possible zero values. The Tweedie distribution has been used mostly for the purpose of insurance claims

analysis, but has seldom been applied in the field of natural hazards, such as HWs mortality (Kim et al., 2017), droughts (Tijdeman et al., 2020), and rainfall analysis (Dunn, 2004; Hasan and Dunn, 2011). The main advantage of the Tweedie distribution is the possibility of considering many distributions for the continuous and semi-continuous domain such as: normal, Gamma, Poisson, Compound Gamma-Poisson, and Inverse Gaussian (Bonat and Kokonendji, 2017; Rahma and Kokonendji, 2021; Shono, 2008; Temple, 2018). Moreover, for some of these distributions (i.e. Poisson mixtures of gamma distributions) it explicitly enables the fitting of zero-inflated data. Tweedie distribution main limitation is the complex distribution's fitting methodology and the difficulties to compare it to other models via information criteria such as the Akaike's information criterion (Shono, 2008).

To:

Instead, for the first time we use a distribution allowing for the direct fitting of zero-values (years with no event): the zero-inflated distribution of Tweedie families (Jorgensen, 1987; Tweedie, 1984). This distribution is also used to estimate HWs and CWs frequency of occurrence. The Tweedie distribution has been used mostly for the purpose of insurance claims analysis, but has seldom been applied in the field of natural hazards, such as HWs mortality (Kim et al., 2017), droughts (Tijdeman et al., 2020), and rainfall analysis (Dunn, 2004; Hasan and Dunn, 2011). The main advantage of the Tweedie distribution is the possibility of considering many distributions for the continuous and semi-continuous domain such as: normal, Gamma, Poisson, Compound Gamma-Poisson, and Inverse Gaussian (Bonat and Kokonendji, 2017; Rahma and Kokonendji, 2021; Shono, 2008; Temple, 2018). Moreover, for some of these distributions (i.e. Poisson mixtures of gamma distributions) it explicitly enables the fitting of zero-inflated data. The distribution's

main limitation is the complex distribution's fitting methodology and difficulties in obtaining its relevant information criteria such as the Akaike's information criterion (Shono, 2008) The implication of these limitations are that the fitting of the Tweedie distribution is computationally intensive and that it is difficult to compare its goodness of fit to other distribution via the information criteria.

• Line 98: What does "temperature vulnerability" mean? Can you specify this in the manuscript?

Thank you for this suggestion and yes indeed this is now specified with the text going from:

Temperature vulnerability has also been appraised at city scale for HWs mortality (Ellena et al., 2020) and at regional scale (López-Bueno et al., 2021) for CWs mortality. Karanja & Kiage (2021) and Cheng et al. (2021) provide an overview of the different types of indicators used in the literature to quantify vulnerability

to:

Vulnerability indicators, in combination with the temperature-mortality relationship, have also been appraised at city scale for HWs (Ellena et al., 2020) and at regional scale (López-Bueno et al., 2021) for CWs. Karanja & Kiage (2021).

• Lines 105-106: It might be worth to name some examples of vulnerability factors used by Frigerio & De Amicis that are also important in the context of your study

Thank you for this comment, indeed and these have been added. The sentence being changed from:

Studies on social vulnerability to natural hazards in Italy used a diversity of indicators derived from the census records (Frigerio and De Amicis, 2016).

To:

A study on social vulnerability to natural hazards in Italy (Frigerio and De Amicis, 2016) used 7 indicators (i.e. family structure, education, socioeconomic status, employment, age, race and ethnicity and population growth) derived from the freely-available census records.

• Lines 125-126: Something seems to be missing here

Thank you for this comment, indeed, you are right:

The sentence has been changed from to:

The latter compared the hospital admissions due to HWs in summer months of three years (2003, 2006, and 2009) possible heat health issues among elder women.

To:

The latter compared the hospital admissions due to HWs in summer months of three years (2003, 2006, and 2009) and found heat health related issues among elderly women.

• Line 135: Might be worth to name the indicators again

Thank you for this suggestion, this has been done by changing the sentence from:

Quantify HWs and CWs hazards and their return level at a very high spatial resolution (250m) by combining for the first time i) the indicators proposed by Russo et al., (2015) and Smid et al., (2019), together with ii) the Tweedie distribution

to:

Quantify HWs and CWs hazards and their return level at a very high spatial resolution (250m) by combining for the first time i) the indicators (HWMId, CWMId)

proposed by Russo et al., (2015) and Smid et al., (2019), together with ii) the Tweedie distribution

• Lines 169-174: To me, this seems to fit better to the discussion or conclusions, as it is an assessment (and opinion) rather than a description of methods.

Thank you for this comment, the authors agree with this and the paragraph in question has been moved and is now the first paragraph of the discussion.

• Line 211: "always <T 0" - maybe a typo?

Thank you for noticing this, the T was indeed a typo and has been removed.

• Lines 341ff: Same text appears twice.

Thank you, the duplicate has been removed.

• Line 344: I think that it should be shortly explained what "robust regression technique" is, and how it differs, e.g., from least-squares regression (see also general comment above)

This has been done. The relevant section has been changed from:

The trends are analyzed using the robust regression technique (Huber, 2011). This method is often used throughout the literature for assessing trends in natural hazards (Formetta and Feyen, 2019 for multiple hazards and Kishore et al., 2022 specifically for HWs).

To:

The trends are analyzed using the robust regression technique (Huber, 2011) which is often used to assess trends in natural hazards (Formetta and Feyen, 2019 for

multiple hazards and Kishore et al., 2022 specifically for HWs). Robust regression seeks to overcome part of the limitations of traditional regression analysis.

For example, the linear regression least squares method is optimal when the regression's assumptions (normal distribution, independence, equal variance) are valid (Filzmoser and Nordhausen, 2021; Khan et al., 2021). This method can be sensitive to outliers or if normality is dissatisfied (Khan et al., 2021; Brossart et al., 2011). The robust regression method is designed to limit the effect that invalid assumptions have on the regression estimates (see Filzmoser and Nordhausen, 2021 and Alma, 2011 for more details).

• Lines 366-368: I do not understand this sentence. Why does the test reveal a significance level? Isn't the significance level chosen by you? And what does the false discovery rate have to do with this? The latter is rather a method to correct for potential overestimations of the number of grid cells classified as statistically significant.

Thank you for this comment, this is indeed not clear. The authors have changed the sentence from:

A KS test (Figure S3 in the supplementary material) shows that the Tweedie distribution provides a good fit for both CWMId and HWMId, with power parameter values between [1,2] for the entire region. The KS goodness of fit test reveals a significance level of $\alpha_{sig}=5\%$ as well as the false discovery rate for the significance level $2\alpha_{sig}$ for any pixel in the region.

To:

The KS tests p-values (Figure S2 in the supplementary material), indicate that the fitting of the Tweedie distribution with power parameter values between [1,2] cannot be rejected for both HWMId and CWMId.

 • Lines 386-395: I am not sure how I should interpret the population numbers (especially the ones above 1 million) given that the analyzed region has a total population of about 1 million. How can the affected population be 5 million? Is it something like person-years, i.e, the sum over all exposed people in all years? This should be specified in the paper.

Thank you for this comment, it is indeed the sum of the people affected over this time frame. This is now specified in the paper and the sentences are changed from:

In total, between 1980 and 2000, in the study region, about 900 000 people were exposed to a 5-year HW event, 250 000 to 10-year HW event, 3million to 5-year CW event and 1.9 million to 10-year CW event. Between 2000 and 2018, the values increased to over 5millions for 5-year HW event and to about 2.5million for 10-year HW event but decreased to 2.4 million for 5-year CW event and to 500 000 for 10-year CW event.

To:

Summing the overall number of people exposed over intervals (i.e. one person can be exposed each year and therefore counted multiple times over the interval), between 1980 and 2000 about 900 000 people were exposed to a 5-year HW event, 250 000 to 10-year HW event, 3million to 5-year CW event and 1.9 million to 10-year CW event. More recently, between 2000 and 2018, the population exposure values increased significantly to over 5 million for 5-year HW event and to about 2.5 million

for 10-year HW event but the numbers decreased for CW events, to 2.4 million for 5-year CW event and to 500 000 for 10-year CW event.

• Figure 3: The title of one subplot is missing

Thank you for noticing this, this has been adjusted.

The previous figure:

[Figure]

The new figure:

[Figure]

• Lines 483-486: I am not totally convinced by this argument given that the highest mountains in that region are not where the strongest HWs occur. I would be more cautious with this statement.

Thank you for this comment.

We agree with the reviewer, and we rewrote the sentence in order to be more cautious in the concept expressed:

The text was nonetheless adjusted to justify this statement from:

The location of our highest increasing trends in HWs events are concordant to those of the higher increase in temperatures found at higher elevations by Acquaotta et al., (2015) in north-west Italy

To:

Figure 3a indicates that a strong increase in heatwave trends is observed in the northwest and the north of our study area. Both areas are at a high elevation (between ~1000m and ~3900m) and one includes the highest mountain in the analyzed area. These results are consistent to those presented by Acquaotta et al., (2015), which found higher increases in temperatures at higher elevations in north-west Italy.

• Lines 492-493: I do not understand this sentence. What is it meant to say?

Thank you for asking, this has been clarified, the sentence has been changed from:

The two driving factors behind the increase in vulnerability (elderly population and isolation) have also been found as some of the main factors for vulnerabilities in other regions of Europe (López-Bueno et al., 2021; Poumadère et al., 2005)

To:

Consistently with previous studies (e.g. López-Bueno et al., 2021; Poumadère et al., 2005), we found that the elderly population and isolation were the indicators most affecting the increase in extreme temperature vulnerability.

• Lines 517ff: I would say that in certain areas also the change in population plays a role according to Figure 6.

Thank you for this comment, this is what the authors meant by demographic changes (ie. an increasing and aging population). This has been specified further in the article, the sentence is changed from:

The changes in CWs risk is mainly explained by the demographic and vulnerability changes, which are increasing in/around urban areas and decreasing elsewhere.

To:

The changes in CWs risk are explained by the demographic (i.e. an increasing and aging population) and vulnerability changes, which are increasing in/around urban areas and decreasing elsewhere.